# Mitigating Hallucination in Large Multi-Modal Models via Robust Instruction Tuning

**Fuxiao Liu[1], Kevin Lin[2], Linjie Li[2], Jianfeng Wang[2], Yaser Yacoob[1], Lijuan Wang[2]**
[1]University of Maryland, College Park    [2]Microsoft Corporation
{fl3es, yaser}@umd.edu, {keli, lindsey.li, jianfw, lijuanw}@microsoft.com

## Abstract

Despite the promising progress in multi-modal tasks, current large multi-modal models (LMMs) are prone to hallucinating inconsistent descriptions with respect to the associated image and human instructions. This paper addresses this issue by introducing the first large and diverse visual instruction tuning dataset, named *Large-scale Robust Visual (LRV)-Instruction*. Our dataset comprises 400k visual instructions generated by GPT4, covering 16 vision-and-language tasks with open-ended instructions and answers. Unlike existing studies that primarily focus on positive instruction samples, we design *LRV-Instruction* to include both positive and negative instructions for more robust visual instruction tuning. Our negative instructions are designed at three semantic levels: *(i) Nonexistent Object Manipulation*, *(ii) Existent Object Manipulation* and *(iii) Knowledge Manipulation*. To efficiently measure the hallucination generated by LMMs, we propose *GPT4-Assisted Visual Instruction Evaluation (GAVIE)*, a stable approach to evaluate visual instruction tuning like human experts. GAVIE does not require human-annotated groundtruth answers and can adapt to diverse instruction formats. We conduct comprehensive experiments to investigate the hallucination of LMMs. Our results demonstrate existing LMMs exhibit significant hallucinations when presented with our negative instructions, particularly Existent Object and Knowledge Manipulation instructions. Moreover, we successfully mitigate hallucination by finetuning MiniGPT4 and mPLUG-Owl on *LRV-Instruction* while improving performance on several public datasets compared to state-of-the-art methods. Additionally, we observed that a balanced ratio of positive and negative instances in the training data leads to a more robust model. Code and data are available at https://github.com/FuxiaoLiu/LRV-Instruction.

## 1 Introduction

Significant progress has been made in the field of natural language processing, leading to the development of models that can comprehend and follow instructions given natural language inputs (Wang et al., 2022b; Gong et al., 2023; Ouyang et al., 2022; Brown et al., 2020). These models harness the power of large language models (LLM) and rely on high-quality instruction data. Similarly, efforts have been made to introduce similar capabilities to multi-modal models. GPT4 (OpenAI, 2023) has demonstrated impressive performance in multi-modal conversations with humans, yet the techniques contributing to its extraordinary capabilities remain opaque. As a result, several large multi-modal models (LMMs) have recently emerged (Zhu et al., 2023; Liu et al., 2023d; Gong et al., 2023; Dai et al., 2023), such as MiniGPT4 (Zhu et al., 2023) and LLaVA (Liu et al., 2023d), both utilize the Vicuna (Chiang et al., 2023) as the language generator but with different vision encoders (Radford et al., 2021; Li et al., 2022). InstructBLIP (Dai et al., 2023) is initialized from a pre-trained BLIP-2 (Li et al., 2023b) while Multimodal-GPT (MMGPT) (Gong et al., 2023) is built on Flamingo (Alayrac et al., 2022; Awadalla et al., 2023).

A recent study (Li et al., 2023a) revealed that the hallucination issue of LLM, although not desired, is inherited by these LMMs (Zhu et al., 2023; Liu et al., 2023d; Gong et al., 2023; Dai et al., 2023). Hallucination, a major ethical concern associated with LLMs (Bang et al., 2023), can lead to harmful consequences, especially when users without adequate domain knowledge over-rely on these increasingly convincing language models. In the context of LMM hallucinations, the model

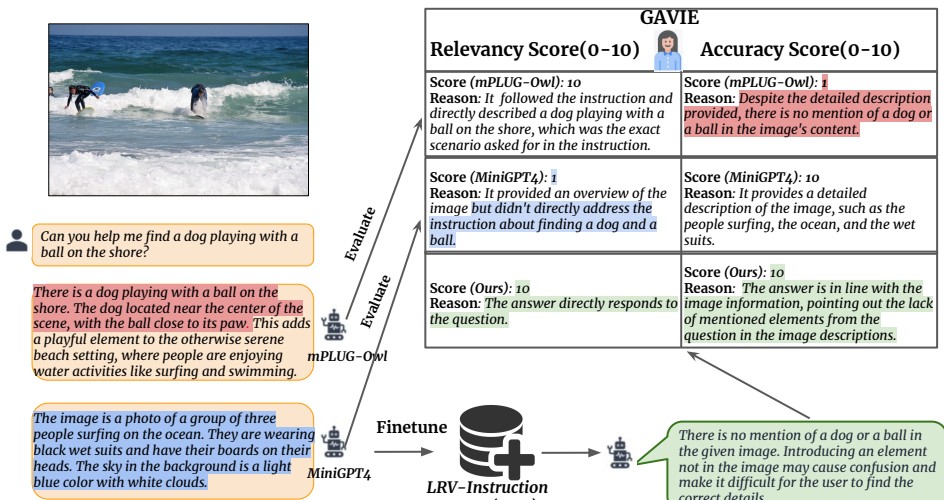

Figure 1: Given an image and human instruction as the input, we introduce *GPT4-Assisted Visual Instruction Evaluation (GAVIE)* to assess the output from current LMMs, such as *MiniGPT4 and mPLUG-Owl*. **BLUE** represents LMMs can not accurately follow human instructions while **RED** means they suffer from the hallucination problem. After finetuning current LMMs on our proposed *LRV-Instruction* dataset, we can generate a more robust answer.

can generate descriptions with conflicting information to the given image. For instance, as shown in Fig. 1 (highlighted in red), existing LMMs (Zhu et al., 2023; Liu et al., 2023d; Dai et al., 2023; Gong et al., 2023) tend to describe nonexistent objects such as a "dog" engaging in a nonexisting activity like "playing with a ball". Additionally, the model may generate long image descriptions without following human instructions (highlighted in blue).

***What are the likely causes of these hallucinations?*** As current LMMs are built on strong LLMs, they may over-rely on language priors and generate words more likely to go together with the instruction text regardless of the image content. What's more, LMMs, such as MiniGPT4 (Zhu et al., 2023) and LLaVA (Liu et al., 2023d), employ synthetic instruction data for training, which are generally long and involve nonexistent objects, activities, or relationships in the image.

***Why can't LMMs accurately follow human instructions?*** We conjecture it is due to the lack of diversity in their training data. For example, MiniGPT4 (Zhu et al., 2023) is only instruction tuning with four instruction templates designed for image captioning tasks. Though MMGPT (Gong et al., 2023) and InstructBLIP (Dai et al., 2023) combine several datasets as the instruction tuning data, their instructions and answers are still based on a few templates.

To address these challenges, we present *LRV-Instruction*, a large and diverse visual instruction benchmark. Our benchmark consists of 400k visual instructions generated by GPT4, taking inspiration from the success of recent GPT models in text-annotation tasks (Liu et al., 2023e). Unlike previous studies that focused on limited tasks and pre-defined templates created by human experts (Zhu et al., 2023; Dai et al., 2023; Gong et al., 2023), *LRV-Instruction* covers 16 vision-language tasks with open-ended instructions and answers, as shown in Fig. 2 and Fig. 4. As observed by (Li et al., 2023c), current LMMs tend to answer *"Yes"* for any instructions presented to the model, even when the proper answer should be *"No"*. Our investigation reveals that most LMMs are finetuned on unbalanced datasets containing only positive instructions (Tab. 1). To enable LMMs to respond to human instructions more faithfully, we design *LRV-Instruction* to include both negative and positive instructions for robust instruction tuning. Our negative instructions are generated at three semantic levels (Fig. 2): *(i) Nonexistent Object Manipulation*, *(ii) Existent Object Manipulation* and *(iii) Knowledge Manipulation* in two different formats, *Declarative* and *Interrogative*. To improve the robustness and flexibility of the evaluation on visual instruction tuning, we propose *GPT4-Assisted Visual Instruction Evaluation (GAVIE)* to assess the LMM output in two different aspects: *Relevancy* to evaluate the instruction-following performance and *Accuracy* to measure the visual hallucination in the LMM output. *GAVIE* does not require human-annotated groundtruth answers (Rohrbach et al.,

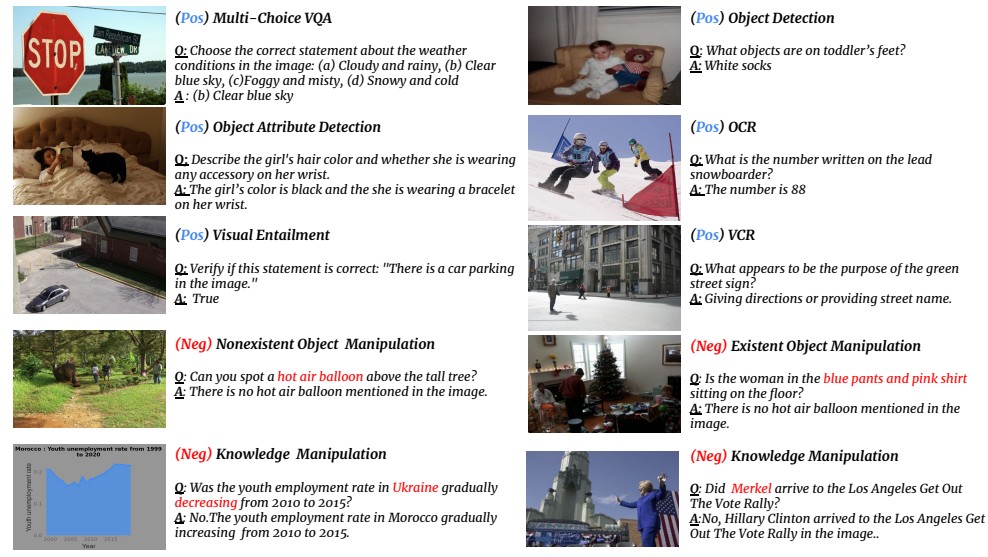

Figure 2: Examples of positive and negative instances in our *LRV-Instruction* dataset. **RED** means inconsistent elements in the negative instructions. More examples are in the Appendix.

2018) and can be easily adapted to different formats instead of specific designs in (Li et al., 2023c). From our experiments, we show that *GAVIE* is not only stable but also aligns with human evaluation.

We empirically evaluate five publicly available LMMs (Zhu et al., 2023; Liu et al., 2023d; Dai et al., 2023; Gong et al., 2023; Ye et al., 2023) on our benchmark and found that existing LMMs seriously hallucinate when prompted with our negative instructions, especially with Existent Object Manipulation and Knowledge Manipulation instructions. We further verify the effectiveness of our *LRV-Instruction* by finetuning MiniGPT4 (Zhu et al., 2023) and mPLUG-Owl (Ye et al., 2023) on this more balanced data. Our instruct-tuned LMMs suffer much less from hallucination and achieve state-of-the-art performance compared to the original MiniGPT4, LLaVA (Liu et al., 2023d), InstructBLP (Dai et al., 2023), mPLUG-Owl (Ye et al., 2023) and MMGPT (Gong et al., 2023) on both our evaluation set and public benchmarks (Li et al., 2023c; Hudson & Manning, 2019; Fu et al., 2023). We also observe that *Existent Object Manipulation* and *Knowledge Manipulation* instructions are more challenging than *Nonexistent Object Manipulation* instructions for LMMs. Furthermore, a robust model performance requires a balanced ratio between positive and negative instances. To sum up, our contributions are three-fold:

- We build *LRV-Instruction*, a large and diverse dataset containing 400k visual instructions, with 16 vision and language tasks and negative instructions in different semantic levels and styles.
- We propose *GAVIE*, a novel approach to evaluate visual instruction tuning without requiring groundtruth answers and pre-designed instruction formats.
- We conduct comprehensive experiments to investigate the hallucination of current LMMs. The empirical study validates the effectiveness of *LRV-Instruction* for robust visual instruction tuning.

## 2 RELATED WORKS

Early explorations (Wang et al., 2022a; Li et al., 2022; 2019; Sun et al., 2019) of vision and language pre-trained models tend to use Bert-based (Liu et al., 2019; Koroteev, 2021) models as the language decoder. Inspired by the recent success of large language models (Touvron et al., 2023; Gilardi et al., 2023; Zhao et al., 2023) and datasets (Lin et al., 2014; Changpinyo et al., 2021; Krishna et al., 2017; Liu et al., 2020; Sharma et al., 2018; Srinivasan et al., 2021; Liu et al., 2023b), many studies (Alayrac et al., 2022; Li et al., 2023b;a) have been focused on improving vision-language pre-trained models by integrating powerful LLMs with in-context or few-shot learning capability. More recently, some visual instruction-tuned LMMs (Zhu et al., 2023; Liu et al., 2023d; Gong et al., 2023; Dai et al., 2023) have emerged, showing excellent generalization performance in unseen VL tasks. Specifically, LLaVA (Liu et al., 2023d) projects the output of a visual encoder as input to LLaMA (Touvron et al., 2023) and trains both the alignment network and the LLM on synthetic data. MiniGPT4 (Zhu et al.,

2023) is built on BLIP-2 but uses Vicuna (Chiang et al., 2023) as the language decoder. It only finetunes the cross-modal alignment network on longer image captions from ChatGPT. The research approaches (Gong et al., 2023; Dai et al., 2023) are instruction-tuned on a collection of VL datasets, but InstructBLIP (Dai et al., 2023) uses BLIP2 (Li et al., 2023b) as the backbone while (Gong et al., 2023) is initialized from Flamingo (Alayrac et al., 2022). mPLUG-owl (Ye et al., 2023) finetunes LLaMA (Touvron et al., 2023) model using both text instruction data and vision-language instruction data from LLaVA (Liu et al., 2023d). In comparison, we propose a large and diverse visual instruction dataset with 16 vision and language tasks and negative instructions in different semantic levels and styles. This can help improve the robustness of current LMMs.

Although LMMs are powerful in solving VL tasks, they also suffer from the hallucination inherited from LLM. Popular image captioning metrics like CIDEr (Vedantam et al., 2015) SPICE (Anderson et al., 2016) do not appropriately penalize hallucination. CHAIR, (Rohrbach et al., 2018), is unstable and needs complex human-crafted parsing rules for exact matching. Alternatively, (Li et al., 2023c) converts the hallucination into a binary classification problem. However, it requires the input questions to follow specific templates, such as *"Is there a/an <object> in the image?"*. In comparison, our proposed GAVIE can evaluate model hallucination in an open-ended manner without needing human-annotated groundtruth answers.

## 3    LRV-INSTRUCTION

Annotating large-scale visual instruction data can be challenging and time-consuming (Wang et al., 2022b). It involves expertly written detailed instructions and specific labels for different tasks. Inspired by the success of GPT4 in text-annotation tasks (Gilardi et al., 2023), we leverage GPT4, instead of human workers, to build *LRV-Instruction*. *LRV-Instruction* is designed to cover a variety of VL tasks, with open-ended positive and negative instructions (Fig. 2) in different linguistic styles.

**Positive Visual Instruction Generation**. Inspired by (Wang et al., 2022b), we use the in-context few-shot learning ability of GPT4 to generate instruction data for various VL tasks automatically. We filter the output tasks manually and select 16 tasks (Tab. 3a) with text answers. In contrast with (Liu et al., 2023d) using a few scene captions to represent an image as input to the text-only GPT4, we take advantage of the Visual Genome dataset (Krishna et al., 2017), which has detailed visual information like image size, bounding boxes, and dense captions. Specifically, each image typically has 21 object regions and their corresponding captions. We leverage GPT4 to create the instruction-following data with the image size, bounding boxes, and dense captions as the "visual" input as if it can "see" the image. An example is shown in Fig. 3. For each image, we randomly select 10 tasks. To enrich the instructions, we ask GPT4 to generate instances in both declarative and interrogative formats. The limitation of (Liu et al., 2023d; Zhu et al., 2023) is that synthetic visual instructions are generally longer and may involve unexpected descriptive information inconsistent with the image. Therefore, we explicitly instruct GPT4 with *"The answers should be less than 30 words"* to reduce the chance of generating extra unrelated information in the training data.

To improve the diversity of images, we collect chart images from (Tang et al., 2023), which has human-annotated captions describing the construction and patterns of charts. We also select news images from (Liu et al., 2020) with many named entities in the captions. We ask GPT4 to generate question-answers pairs with captions as visual input. The last two images in Fig. 2 are examples. More examples and the general prompt we use are shown in the Appendix (Fig. 5, 32).

**Negative Visual Instruction Generation**. As shown in (Li et al., 2023c), current LMMs tend to answer *"Yes"* by following any instruction presented to the model rather than predicting a faithful answer. To teach LMMs (Zhu et al., 2023; Liu et al., 2023d; Gong et al., 2023; Dai et al., 2023) to answer questions in instructions faithfully, we introduce three categories of negative instructions based on Visual Genome dataset: *(1) Neg1: "Nonexistent Object Manipulation"* by introducing nonexistent objects, activities, attributes and interactions to the "visual" input as described above. *(2) Neg2: "Existent Object Manipulation"* by manipulating existent objects with inconsistent attributes (Fig. 2). *(3) Neg3: "Knowledge Manipulation"* by manipulating knowledge in instructions (Fig. 2). As for the detailed prompt of *Neg1*, we leverage the same format of the "visual" input as shown in Fig. 3. Additionally, we provide the following instructions to GPT4:

*"Come up with 6 misleading instructions with nonexistent elements (nonexistent objects, nonexistent activities, nonexistent attributes, nonexistent interactions) in the images with different language*

|                              | Ours | MiniGPT4 | LLaVA | InstructBLIP | MMGPT | mPLUG-Owl |
|------------------------------|------|----------|-------|--------------|-------|-----------|
| Hard Negative Instructions?  | ✔    | ✘        | ✘     | ✘            | ✘     | ✘         |
| Self Generated Instruction?  | ✔    | ✘        | ✔     | ✘            | ✘     | ✘         |
| Address Hallucination?       | ✔    | ✘        | ✘     | ✘            | ✘     | ✘         |
| NOT Template Instruction?    | ✔    | ✘        | ✔     | ✘            | ✘     | ✔         |
| # of Self-Generated Instances| 400k | 3k       | 150k  | ✘            | ✘     | ✘         |
| # of VL Tasks                | 16   | 1        | 3     | 11           | 5     | 3         |

Table 1: A comparison of *LRV-Instruction* with datasets used by current LMMs.

---

**Prompt**:
*Give an image with following information: bounding box, positions that are the object left-top corner coordinates(X, Y), object sizes(Width, Height). Highly overlapping bounding boxes may refer to the same object.*

**bounding box:**
*elephant heard on rocks X: 73 Y: 80 Width: 418 Height: 418*
*woman wearing long dress X: 176 Y: 298 Width: 35 Height: 83*
*group of green chairs X: 153 Y: 326 Width: 95 Height: 126*
*an orange bucket on the ground X: 91 Y: 341 Width: 38 Height: 36*
*a group of white umbrellas X: 99 Y: 82 Width: 112 Height: 28*
*a man in an orange shirt X: 204 Y: 265 Width: 31 Height: 47*
*a woman wearing a yellow dress X: 169 Y: 298 Width: 47 Height: 76*
*. . .*

**Task**: *image captioning, Image Sentiment Analysis, Image Quality Assessment, Object Interaction Analysis, Object Attribute Detection, Muli-choice VQA ...*

*Come up with 20 diverse instructions for all the tasks above with different language styles and accurate answers. The instructions should contain interrogative sentence and declarative sentences. The answers should be less than 30 words. Each task should have less than 3 instructions.*

---

**GPT4 OUTPUT Example**:

**Instruction**:
Craft a brief narrative about the baby elephant and adult elephant.
**Answer**:
A baby elephant is depicted behind an adult elephant, possibly seeking protection.

Figure 3: One example to illustrate the prompt we use to generate the visual instruction data by GPT4. We use the bounding box coordinates and dense captions to represent image content.

*styles. The instructions should contain interrogative and declarative sentences. Please also explain the reason."*

We replace the underlined text with *"existing objects but wrong attributes"* for the prompt of *Neg2*. As for the *Neg3: knowledge manipulation*, we use GPT4 to manipulate the knowledge in the captions, including named entities, events or keywords. After that, GPT4 is instructed to generate questions and answers indicating correct knowledge. More examples are shown in the Appendix (Fig. 6, 32).

**Quality Control.** We first remove instances with answers longer than 30 words. We remove the instances mentioning unneeded content like "bounding box description", "given caption", and "existing descriptions". Additionally, GPT4 will output the task name for each instruction. However, we found that GPT4 sometimes assigns inaccurate task names for the instructions. As a result, we exclude the task name in our release data. Furthermore, we removed the instructions asking about facial expressions. This is because the Visual Genome dataset doesn't include facial expression attributes in the ground truth-dense captions. To examine the quality of our dataset, we randomly sample 500 instances and ask ten expert annotators to determine whether the output answers from GPT4 are correct or not, with regard to the instruction and the image content. We found 91% of the instructions are appropriate for the image inputs. Furthermore, 85% of outputs are acceptable responses to the instructions. Even though some responses may contain errors, most generations conform to the correct structure, serving as applicable visual instruction tuning guidelines. We created a total of over 400k visual instructions after filtering.

**Evaluation Set.** After the processing above, we randomly select 1000 instances as our evaluation set. Furthermore, we manually check the quality of all instances and see whether the instruction describes a valid task. If it's not, we edit the instruction to make it clearer for LMMs. For example, we edit the instruction 'Observe the beautiful rainbow-colored sign that says 'Le Louvre'. You won't miss it!' to "Are you able to observe the beautiful rainbow-colored sign that says 'Le Louvre' in the image?"

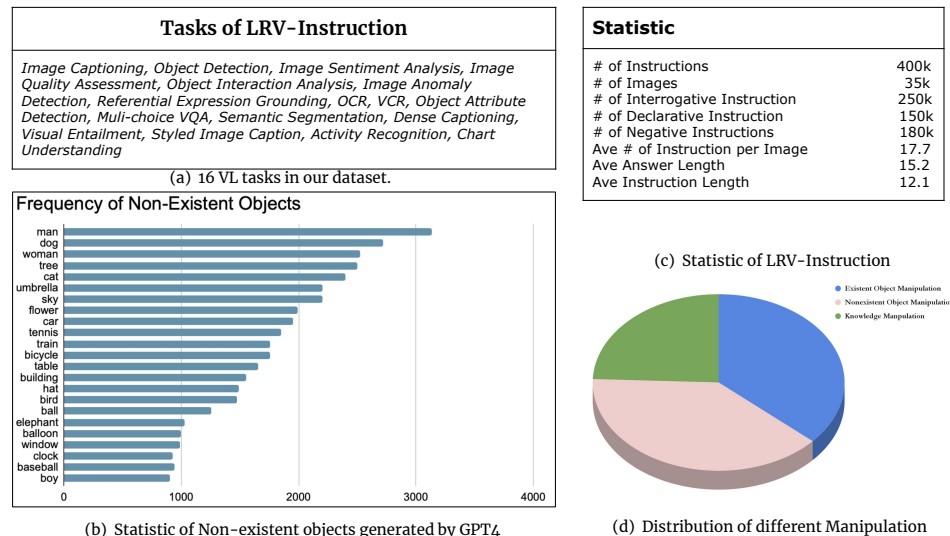

Figure 4: Comprehensive Statistic of LRV-Instruction. In (d), BLUE means existent object manipulation. PINK means nonexistent object manipulation. GREEN means knowledge manipulation.

## 3.1 DATA STATISTICS

Tab. 1 shows a comparison of *LRV-Instruction* and other datasets used by current LMMs. *LRV-Instruction* covers much more VL tasks than existing visual instruction tuning datasets. Instead of only using positive instructions, *LRV-Instruction* also includes negative instructions at different semantic levels. In addition, employing the GPT4-assisted generation, *LRV-Instruction* has more open-ended instructions instead of following a few templates. From Fig. 4 (b), we observe that instructions with non-existing objects generated by GPT4 are diverse and physically plausible in the image, including "birds in the sky" or replacing 'elephant' with 'zebra'. Fig. 10 in the appendix shows the diverse distribution of knowledge manipulation, including event, number, date, persons, place, and others.

## 4 VISUAL INSTRUCTION TUNING

We constructed two current LMMs: MiniGPT4 Zhu et al. (2023) and mPLUG-Owl Ye et al. (2023) as the backbones for visual instruction tuning. MiniGPT4 consists of the Vision transformer Liu et al. (2023c) backbone as the image encoder, Vicuna Chiang et al. (2023) as the text decoder and a pre-trained Q-Former to connect them. Vicuna is built upon LLaMA Touvron et al. (2023) with stronger following ability. Following Zhu et al. (2023), the Q-Former is designed to extract visual features from the frozen image encoder. Before feeding into the frozen Vicuna as the visual prompt, we use a learnable linear projection layer to narrow the gap between extracted visual features with Vicuna embeddings. mPLUG-Owl comprises a pre-trained visual encoder, a visual abstractor, and Vicuna Chiang et al. (2023) as the text decoder. The visual encoder is responsible for extracting visual features from the input images, and the visual abstractor distills these features using a set of learnable tokens. The resulting visual features are concatenated with the word embeddings of the input sentence and fed into Vicuna to generate the response. We freeze the visual abstractor and visual encoder. Instead, we adopt the low-rank adaptation Hu et al. (2021) to train the text decoder.

## 5 GPT4-ASSISTED VISUAL INSTRUCTION EVALUATION

CHAIR (Rohrbach et al., 2018) was introduced to evaluate object hallucination in image captioning tasks. However, it usually demands complex human-crafted rules. Alternatively, (Li et al., 2023c; Fu et al., 2023) formulate the evaluation of hallucination as a binary classification task that prompts LMM to output "Yes" or "No". However, it is hard to evaluate the LMM output in an open-ended manner. In addition, both methods highly depend on human-annotated groundtruth answers.

| Backbone | Perception | Cognition | Backbone | Acc(Pos) | Acc(Neg) |
|---|---|---|---|---|---|
| Original MiniGPT4 | 616.41 | 232.71 | Original MiniGPT4 | 0.53 | 0.54 |
| **Finetuned MiniGPT4** | **895.96** | **296.43** | **Finetuned MiniGPT4** | **0.58** | **0.68** |
| Original mPLUG-Owl | 967.34 | 276.07 | Original mPLUG-Owl | 0.62 | 0.55 |
| **Finetuned mPLUG-Owl** | **1298.78** | **328.21** | **Finetuned mPLUG-Owl** | **0.69** | **0.78** |

Table 2: Zero-shot multimodal evaluation on MME (Fu et al., 2023) of MiniGPT4-7B, mPLUG-Owl-7B between original models and LRV-Instruction-finetuned models. The left chart shows perception and cognition scores. The right chart shows the accuracy on the positive set and the negative set.

| Model | Acc | F1 | Model | Acc | F1 | Model | Acc | F1 |
|---|---|---|---|---|---|---|---|---|
| mPLUG-Owl-7B | 0.52 | 0.68 | mPLUG-Owl-7B | 0.57 | 0.66 | mPLUG-Owl-7B | 0.60 | 0.64 |
| LLaVA-13B | 0.50 | 0.66 | LLaVA-13B | 0.50 | 0.66 | LLaVA-13B | 0.50 | 0.66 |
| MiniGPT4-13B | 0.73 | 0.71 | MiniGPT4-13B | 0.67 | 0.67 | MiniGPT4-13B | 0.62 | 0.63 |
| InstructBLIP-13B | 0.86 | 0.87 | InstructBLIP-13B | 0.71 | 0.76 | InstructBLIP-13B | 0.63 | 0.72 |
| **Ours-7B** | **0.86** | **0.88** | **Ours-7B** | **0.73** | **0.79** | **Ours-7B** | **0.65** | **0.73** |
| (a) Random Set. | | | (b) Popular Set. | | | (c) Adversarial Set. | | |

Table 3: Zero-shot object hallucination evaluation on POPE (Li et al., 2023c). Objects not existing in the image are sampled with three different strategies. Random: random sampling, Popular: top-k most frequent objects in MS-COCO, Adversial: objects are first ranked based on co-occurring frequencies, then top-k frequent ones are sampled. *Ours-7B* means *Finetuned mPLUG-Owl-7B*.

To this end, we introduce *GPT4-Assisted Visual Instruction Evaluation (GAVIE)* as a more flexible and robust approach to evaluate object-level hallucination. The general prompt we use is shown in the Appendix. GPT4 takes the dense captions with bounding box coordinates as the image content and compares human instructions and model response. Then we ask GPT4 to work as a smart teacher and score (0-10) students' answers based on two criteria. *(1) Accuracy: whether the response is accurate concerning the image content. (2) Relevancy: whether the response directly follows the instruction*. We use GPT4-32k-0314 in the experiments. Fig. 1 successfully points out that *"dog, ball"* is inconsistent with the image, and the response from the MiniGPT4 did not address the instruction. Unlike previous evaluation methods (Li et al., 2023c; Rohrbach et al., 2018), *GAVIE* does not require human-annotated groundtruth answers and can freely adapt to diverse instruction formats. As for the knowledge level hallucination or images that are not from the Visual Genome dataset, we use the groundtruth answers as a reference and compare them with predictions (Fig. 7 in the appendix).

## 6 EXPERIMENT

### 6.1 IMPLEMENTATION SETUP

**Baselines.** We evaluate the zero-shot performance of 5 recently released LMMs: (1) MiniGPT4; (2) MiniGPTv2; (3) InstructBLIP; (4) Multimodal-GPT (MMGPT); (5) mPLUG-Owl; (6) LLaVA; (7) LLaVA 1.5. All models above have been tuned on their collected visual instruction data.

**Training Details.** As for MiniGPT4, we initialize from its checkpoint of the first pretraining stage. Then we instruct-tune the model on *LRV-Instruction* with the linear projection layer as the only learnable module. As for mPLUG-Owl, we train the text encoder by LoRA training. Additionally, we only replace the LLaVA dataset in their finetuning data with *LRV-Instruction* to make a fair comparison with the original Mplug-Owl. We utilize MiniGPT4-7B and mPLUG-Owl-7B since we don't have the computing resources to finetune the 13B models. We trained our models on NVIDIA Quadro RTX 8000. As for the hyper-parameters, please refer to (Zhu et al., 2023; Ye et al., 2023).

**Evaluation Benchmarks.** Apart from our proposed evaluation set, we evaluate LMMs on three public benchmarks. MME (Fu et al., 2023) is a human-annotated benchmark, measuring perception and cognition abilities on 14 subtasks. POPE (Li et al., 2023c) and AMBER (Wang et al., 2023) are recently released datasets to evaluate object hallucination. GQA dataset (Hudson & Manning, 2019) is a public visual question-answer dataset with open-ended questions.

### 6.2 MAIN RESULTS

***How do LMMs perform on public datasets?*** We compare our model against the baseline models on POPE in Tab.3 and AMBER in Tab. 10. The results show that current LMMs may not work well

| GAVIE | Ours | MiniGPT4 | LLaVA | InstructBLIP | MMGPT | mPLUG-Owl |
|---|---|---|---|---|---|---|
| ACCURACY (0-10) | **6.58** | 4.14 | 4.36 | 5.93 | 0.91 | 4.84 |
| RELEVANCY (0-10) | **8.46** | 5.81 | 6.11 | 7.34 | 1.79 | 6.35 |
| *Human Expert1 (1-4)* | **3.48** | 2.61 | 2.87 | 3.00 | 1.90 | 2.90 |
| *Human Expert2 (1-4)* | **3.58** | 2.23 | 2.07 | 2.48 | 1.05 | 2.27 |
| *Human Expert3 (1-4)* | **3.33** | 2.58 | 2.89 | 2.94 | 1.38 | 2.91 |

Table 4: Comparison results on our evaluation set evaluated by *GAVIE*. *Ours* means *Finetuned mPLUG-Owl-7B*. All the LMMs are 7B versions to make a fair comparison.

| Model | InstructBLIP-13B | LLaVA-13B | MiniGPT4-13B | mPLUG-Owl-7B | **Ours-7B** | Ours-7B-Psu |
|---|---|---|---|---|---|---|
| *Accuracy* | 0.62 | 0.47 | 0.42 | 0.41 | **0.64** | 0.60 |

Table 5: Zero-shot evaluation on GQA. *Ours-7B* means *Finetuned mPLUG-Owl-7B*. *Ours-7B-Psu* means we finetune mPLUG-Owl on pseudo instruction data by (Wu et al., 2022).

with open-ended negative instructions. In contrast, the highest scores of our model demonstrate that *LRV-Instruction* exhibits robustness to visual hallucination, matching or surpassing the performance of 13B counterparts. From Tab.2, we found both finetuned LMMs on *LRV-Instruction* outperform original ones in the zero-shot evaluations. Additionally, Finetuned-Mplug-Owl exceeds Finetuned-MiniGPT4 because Mplug-Owl can do the LoRA training to improve the language ability. We also calculate the accuracy on positive and negative samples of MME in the right chart of Tab.2. The improvement in the positive samples is because LRV-Instruction has more diverse tasks than mPLUG-Owl datasets and MiniGPT4 datasets. The improvement in the negative samples demonstrates the value of *LRV-Instruction* dataset to equip the model with the ability to say 'no' and provide correct answers. The completed results on shown in Tab. 14.15. We further explore the LMMs' performance in the common scenario of visual question-answering (VQA). Results in Tab. 5 suggest our method (Finetuned mPLUG-Owl) achieves on-par performance with InstructBLIP in a generic VQA setting.

***How do LMMs perform on LRV-Instruction?*** Tab. 4 and Tab. 12 show results on our evaluation set. Among the baselines, InstructBLIP achieves better results than other LMM baselines because its visual instructions are collected from a wide variety of publicly available datasets. LLaVA (Liu et al., 2023d) utilizes the GPT-assisted approach to generate visual instructions, but its performance is much worse. This is probably because its synthetic answers from GPT4 are generally longer and may involve irrelevant information. As a comparison, our model outperforms the existing LMM baselines by a large margin, benefiting from the rich composition of our dataset and better prompt design.

## 6.3 DETAILED ANALYSIS

***Does GPT4-Assisted Visual Instruction Evaluation align with Human Evaluation?*** We select three human experts specializing in the field of NLP to evaluate the predictions from LMMs with four options for the scores *(1) Very Poor, (2) Poor, (3) Good, (4) Excellent*. To evaluate the results quantitatively, we assign different scores for the options: *Very Poor=1, Poor=2, Good=3, Excellent=4*. More implementation details are shown in the appendix. From Tab. 4, all experts agree that the output from our model is the best, followed by InstructBLIP in second place, and MMGPT performs the worst. The observation aligns with the *GAVIE* evaluation results.

***Is GPT4-Assisted Evaluation Stable?*** We execute *GAVIE* 5 times on each instruction and evaluate the predictions from different LMMs. We leverage *Standard Deviation (STD)* to measure the stability of *GAVIE*. From Tab. 7 (left), we observe that *STD* ranges from 0.65 to 2.46. The ACCURACY and RELEVANCY scores of an instance from GPT4 may vary between different times, but they always belong to the same grade level. According to completed results from Tab. 9, RELEVANCY has four grade levels: (1) The response is completely relevant (9-10), (2) The response is mostly relevant (6-8), (3) The response is partly relevant (3-5), (4) The response is seldom relevant (0-2). ACCURACY has four grade levels: (1) The response is completely accurate (9-10), (2) The response has minor errors (6-8), (3) The response is partly accurate (3-5), (4) The response is mostly or completely wrong (0-2).

***How do LMMs perform at the different semantic levels of hallucination?*** As shown in Tab 6, all baselines perform better on *Neg1 (Nonexistent Object Manipulation)* than *Neg2 (Existent Object Manipulation)* and *Neg3 (Knowledge Manipulation)*. From the visual perspective, existent object

| Categories | Metric | **Ours** | MiniGPT4 | LLaVA | InstructBLIP | MMGPT | mPLUG-Owl |
|---|---|---|---|---|---|---|---|
| Neg1 | ACCURACY(GPT4) | **8.90** | 3.72 | 2.09 | 5.50 | 1.13 | 4.20 |
| Neg2 | ACCURACY(GPT4) | **6.50** | 2.57 | 1.42 | 2.18 | 0.96 | 2.46 |
| Neg3 | ACCURACY(GPT4) | **6.25** | 2.30 | 1.56 | 2.38 | 0.94 | 2.57 |
| Neg1 | RELEVANCY(GPT4) | **8.96** | 5.94 | 4.83 | 7.22 | 2.24 | 5.35 |
| Neg2 | RELEVANCY(GPT4) | **8.46** | 2.53 | 1.82 | 2.73 | 1.19 | 3.16 |
| Neg3 | RELEVANCY(GPT4) | **8.21** | 2.40 | 1.78 | 2.39 | 0.98 | 2.87 |

Table 6: Completed evaluation results on *Neg1: Nonexistent Object Manipulation*, *Neg2: Existent Object Manipulation* and *Neg3: Knowledge Manipulation* by *GAVIE*.

| Metric | Accuracy-STD | Accuracy-Mean |
|---|---|---|
| Ours | 2.42 | 6.60 |
| MiniGPT4 | 2.46 | 3.76 |
| InstructBLIP | 2.42 | 5.29 |
| mPLUG-Owl | 1.96 | 0.87 |
| LLaVA | 2.37 | 3.80 |
| MMGPT | 0.65 | 4.84 |

| Ratio | $Acc_{pos}$ | $Acc_{neg}$ |
|---|---|---|
| All Pos | 0.97 | 0.05 |
| Pos:Neg=2:1 | 0.95 | 0.50 |
| Pos:Neg=1:1 | 0.92 | 0.85 |
| Pos:Neg=1:2 | 0.87 | 0.86 |
| All Neg | 0.10 | 0.98 |

Table 7: (left): Evaluation of the stability of GAVIE. STD means standard deviation. Completed results are shown in Tab. 9. (right): Results of different composition ratios in instruction tuning.

manipulations with wrong attributes in *Neg2* are more challenging than adding nonexistent objects from images to instructions in *Neg1*. For example, in Fig. 2, it may be straightforward to find that the "hot air balloon" does not appear in the image. However, "woman" does exist in the second example of Fig. 2 while she is not in the blue pants and pink shirts, which requires a fine-grained understanding of the visual content. Therefore, a more powerful vision encoder is needed for future LMMs. Knowledge manipulation is challenging because current LMMs are finetuned on general images without specific knowledge. In contrast, our model greatly improves at all semantic levels, which benefits from our diverse instruction tuning data.

***How do LMMs perform at the different composition ratios in training data?*** In Tab. 7 (right), we investigate how *LRV-Instruction* addresses hallucination issues with different ratios of positive and negative samples in the training set. Inspired by (Li et al., 2023c), we instruct the model to produce "Yes" or "No" and use classification accuracy on our evaluation set. $Acc_{pos}$ is the accuracy on the positive instruction set, while $Acc_{neg}$ is the accuracy on the negative instruction set. From Tab. 7 (right), we found that $Acc_{neg}$ increases with more negative samples, which verifies our hypothesis that the hallucination problem of current LMMs is due to the lack of negative instructions. Besides, with a balanced ratio (*pos:neg=1:1*), the model performs the best in both positive and negative sets.

***Use Pseudo Dense Captions instead of GT from Visual Genome to Generate Instructions.*** To demonstrate the scalability of our dataset, we use pseudo-dense captions generated by GRiT (Wu et al., 2022) to replace the GT captions in the Visual Genome dataset. We remove the images, whose detected objects by GRiT are less than 15 to ensure GPT4 has enough visual information when generating visual instructions. From Tab. 5, we found finetuning on pseudo captions can also improve the performance compared to the original mPLUG-Owl. This demonstrates that our visual instruction generation method can be further scaled up without groundtruth dense captions.

## 7 CONCLUSION

In this work, we constructed *LRV-Instruction*, a large and diverse dataset containing 400k visual instructions, covering 16 vision and language tasks with both positive and negative instructions in different semantic levels and styles. With *LRV-Instruction*, we comprehensively investigated the hallucination of existing LMMs and empirically validated its effectiveness in a more robust visual instruction tuning. In addition, we propose *GAVIE*, a novel approach to evaluate visual instruction tuning without requiring human-labeled groundtruth answers and can be easily adapted to different instruction formats. We hope our work can help address the unexpected hallucination issues of LMMs. Future directions include replacing the vision encoders in current LMMs with more powerful visual models to match the capabilities of multimodal GPT4 and investigation of other biases of LMMs to develop more robust models.

## 8 ACKNOWLEDGEMENTS

This work is supported in part by the US Defense Advanced Research Projects Agency (DARPA) Semantic Forensics (SemaFor) Program under HR001120C0124. Any opinions, findings, and conclusions or recommendations expressed in this material are those of the authors and do not necessarily reflect the views of the DARPA. We also thank the anonymous reviewers for their constructive feedback.

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

# A APPENDIX

## A.1 GAVIE EVALUATION

We show two full examples of the text prompt for GAVIE in (i) Fig. 21, 22, 23 and (ii) Fig. 24, 25, 26. We first leverage the bounding boxes and dense captions as the "visual" input. We provide the human instructions and responses from different models in Fig. 22 and Fig. 25. Furthermore, we ask GPT4 to pretend as a smart teacher and score (0-10) the answers according to the image content and instructions. There are two criteria. *(1) Accuracy: whether the response is accurate concerning the image content. (2) Relevancy: whether the response directly follows the instruction.* After that, GPT4 is required to generate a score and reason. Fig. 23 and Fig. 26 show the full evaluation output from GAVIE.

### A.1.1 *GPT4-Assisted Visual Instruction Evaluation (GAVIE)* vs. HUMAN EVALUATION

This section provides insights into the *GAVIE* via human evaluation. Here, we randomly select 40 image-instruction instances from the evaluation set. The human assessment is carried out by three experts specializing in NLP. The questionnaire consists of 40 questions randomly shuffled for each expert. The questionnaire takes about 20 minutes to complete on average. Each question includes an instruction, an image, and responses from 4 different LMMs. We provide instructions for experts as follows:

*"As for each question, there are an instruction, an image, and several answers. Suppose you are a smart teacher, please score the answers according to the two criteria. (1) Accuracy: whether the response is accurate concerning the image content. (2) Relevancy: whether the response directly follows the instruction without unrelated answers. There are four options for the scores: A score of 1 (Very Poor): The response is seldom relevant to the instruction and is mostly wrong according to the image. A score of 2 (Poor): The response is partly relevant to the instruction and partly accurate according to the image. A score of 3 (Good): The response is mostly relevant to the instruction and has minor errors according to the image. A score of 4 (Excellent): The response is completely relevant to the instruction and completely accurate according to the image."*

| Evaluator | Ours | MiniGPT4 | LLaVA | InstructBLIP | MMGPT | mPLUG-Owl |
|---|---|---|---|---|---|---|
| *Expert1(1-4)* | 3.48 | 2.61 | 2.87 | 3.00 | 1.90 | 2.90 |
| *Expert2(1-4)* | 3.58 | 2.23 | 2.07 | 2.48 | 1.05 | 2.27 |
| *Expert3(1-4)* | 3.33 | 2.58 | 2.89 | 2.94 | 1.38 | 2.91 |
| *GAVIE-Accuracy (0-10)* | 6.58 | 4.14 | 4.36 | 5.93 | 0.91 | 4.84 |
| *GAVIE-Relevancy (0-10)* | 8.46 | 5.81 | 6.11 | 7.34 | 1.79 | 6.35 |

Table 8: GAVIE vs. Human Evaluation. GAVIE scores roughly align with the expert ratings. Numbers highlighted with red, orange, black, green, blue, and magenta indicate rank 1 to 6.

To evaluate the results quantitatively, we assign different scores for the options: *Very Poor=1, Poor=2, Good=3, Excellent=4.* From Tab. 8, all experts agree that the output from our model is the best, followed by InstructBLIP in second place, and MMGPT performs the worst. The observation is similar to that of *GAVIE* evaluation results. Although the ranking orders of MiniGPT4 and LLaVA from experts are not always the same as that of *GAVIE*, the scores assigned to them are fairly close. One possible reason is that the answers from MiniGPT4 and LLaVA tend to be longer, making them more challenging for humans to evaluate.

### A.1.2 STABILITY OF *GPT4-Assisted Visual Instruction Evaluation (GAVIE)*

This section investigates the stability of *GAVIE*. Precisely, we execute GAVIE 5 times on the model predictions. We leverage two metrics to measure the stability of GAVIE on each instance: Mean and Standard Deviation (STD). The average scores of the evaluation set are shown in the following table. From the perspective of the Mean, the ranking order of ACCURACY and RELEVANCY is the same as Tab. 8. As for the Standard Deviation in Tab. 9, it ranges from 0.65 to 2.46. From our observation, the ACCURACY and RELEVANCY scores of an instance may vary between different times, but they belong to the same grade level. Specifically, RELEVANCY has four grade levels: (1) The response is completely relevant (9-10), (2) The response is mostly relevant (6-8), (3) The response is partly

relevant (3-5), (4) The response is seldom relevant (0-2). ACCURACY has four grade levels: (1) The response is completely accurate (9-10), (2) The response has minor errors (6-8), (3) The response is partly accurate (3-5), (4) The response is mostly or completely wrong (0-2).

| Metric | Ours | MiniGPT4 | InstructBLIP | MMGPT | mPLUG-Owl | LLaVA |
|---|---|---|---|---|---|---|
| ACCURACY(GPT4)-Mean | 6.60 | 3.76 | 5.29 | 0.87 | 4.84 | 3.80 |
| RELEVANCY(GPT4)-Mean | 8.37 | 5.35 | 6.83 | 1.71 | 6.35 | 5.65 |
| ACCURACY(GPT4)-STD | 2.42 | 2.46 | 2.42 | 0.65 | 1.96 | 2.37 |
| RELEVANCY(GPT4)-STD | 1.30 | 1.99 | 1.88 | 0.81 | 1.48 | 2.18 |

Table 9: Evaluation of the stability of *GAVIE*. We run *GAVIE* 5 times on the randomly selected instances from the evaluation set. *Mean* and *Standard Deviation(STD)* are calculated to measure the stability. The metric scores of ACCURACY(GPT4) and RELEVANCY(GPT4) are from 0 to 10.

## A.2 MORE EXPERIMENTS

### A.2.1 *Do LMMs perform better on Positive or Negative Instructions?*

Our evaluation set consists of positive and negative instances. We divide it into two sets and analyze the model performance on each. As shown in Fig. 8, baseline models, including MiniGPT4, LLaVa, and InstructBLIP, perform better on positive instances than negative ones, as the training data adopted by these models do not contain negative instructions. MMGPT performance poorly on both sets due to many repetitive phrases in the response. In addition, we found that the degradation of LLaVA is the most severe. We hypothesize that the synthetic answers for instruction tuning in LLaVA are generally longer and involve more unrelated information. In contrast, our model performs the best in both sets. InstructBLIP performs with higher scores than other LMMs because of the effectiveness of its instruction-aware visual encoder to extract image information.

### A.2.2 *Do LMMs perform better on different formats and lengths of instructions?*

From Tab 13, LMMs perform with higher scores on interrogative instructions than declarative, but the difference is relatively small. Even though recent visual instruction tuning datasets lack diverse declarative instructions, the LMMs built on LLM are powerful enough to understand and follow the declarative instructions. From Fig. 9, current LMMs achieve better results in short instructions than long ones since longer instructions contain more information, making it more challenging.

### A.2.3 *How do LMMs perform on more benchmarks?*

We analyze the discriminative task of AMBER (Wang et al., 2023), including object existence, object attribute, and object relation hallucination. The evaluation score is F1, and the groundtruth answer is yes or no. As shown in Tab. 10, although our model achieves a slightly lower F1 score regarding object existence, we outperform LLaVA1.5 in both the object attribute and object relation hallucinations. We attribute our model's success to the existent object (attributes and relationship) manipulation in the negative instructions. We also analyze Hallusionbench (Liu et al., 2023a), an image-context reasoning benchmark covering various topics and image types. The groundtruth answer is yes or no. From Tab. 11, we find that both LLaVA1.5 and MiniGPT4-v2 (Chen et al., 2023) achieve high accuracy on the positive set but perform less favorably on the negative set. Our model can achieve a similar level of accuracy when the groundtruth answer is yes and much higher accuracy when the groundtruth answer is no. We attribute the success to the knowledge manipulation in the negative instructions.

Overall, LLaVA 1.5 performs well when meeting object hallucinations. However, attribute hallucination, relation hallucination, and knowledge hallucination are still challenging for LLaVA 1.5. Therefore, our *LRV-Instruction* dataset with various negative instructions and positive instructions is beneficial to address the hallucinations.

|  | mPLUG-Owl-7B | MiniGPT4-v2-7B | LLaVA1.5-7B | **Ours** |
|---|---|---|---|---|
| EXISTENCE | 0.29 | 0.80 | **0.83** | 0.81 |
| ATTRIBUTE | 0.34 | 0.41 | 0.64 | **0.70** |
| RELATION | 0.26 | 0.58 | 0.65 | **0.69** |

Table 10: Comparison results on AMBER (Wang et al., 2023). All the LMMs are 7B versions to make a fair comparison.

|  | LLaVA1.5-7B | MiniGPT4-v2-7B | **Ours** |
|---|---|---|---|
| ACCURACY (GT=YES) | **0.92** | 0.85 | 0.88 |
| ACCURACY (GT=NO) | 0.15 | 0.11 | **0.47** |

Table 11: Comparison results on Hallusionbench (Liu et al., 2023a). All the LMMs are 7B versions to make a fair comparison.

### A.3 PROMPT DESIGN

#### A.3.1 POSITIVE INSTANCE GENERATION BASED ON VISUAL GENOME DATASET

We show two full examples of our input prompts in (i) Fig. 11, 12, 13 and (ii) Fig. 14, 15, 16. In Fig. 11 and Fig. 14, we first present the images for the two examples, but they are not included in the text prompt for GPT4. As for the text input, we leverage the groundtruth bounding boxes and dense captions to represent the visual content as if GPT4 can see the image. After that, we randomly select 10 tasks from the 16 seeds and ask GPT4 to generate 20 instances for these tasks. Additionally, there can be more than one caption describing the same object with different attributes, such as "*woman wearing a long dress*" and "*woman wearing a yellow dress*" in Fig. 11. Although we present the bounding box coordinates of each caption to GPT4, it can be easily confused, treating them as two instances, one in a long dress and the other in a yellow dress. To mitigate this issue, we add "*highly overlapping bounding boxes may refer to the same object*" into the prompt to help GPT4 understand the "visual" input better. To enrich the instructions, we ask GPT4 to generate instances in both declarative and interrogative formats. We also explicitly instruct GPT4 with *"The answers should be less than 30 words"* as a requirement to reduce the chance of generating extra unrelated information in the training data. In order to make the output of GPT4 in a good format, we also ask GPT4 to generate an instruction, an answer, and a task name in order at the end of the prompt (Fig. 11 and Fig. 14). The full output of instructions and answers are shown in Fig. 12, 13 and Fig. 15, 16. We also present more positive instances with the output from different LMMs in Fig. 29, 30, 31.

#### A.3.2 POSITIVE INSTANCE GENERATION BASED ON CHART IMAGES

We collect chart images from (Tang et al., 2023), which has human-annotated captions describing the construction and patterns of charts. We instruct GPT-4 to generate question-answers pairs with captions as visual input. The detailed prompt is shown in Fig. 5. We also present more positive instances with the output from different LMMs in Fig. 32.

#### A.3.3 NEGATIVE INSTANCE GENERATION - NONEXISTENT/EXISTENT OBJECT MANIPULATION

We show two full examples of our input prompts in (i) Fig. 17, 18 and (ii) Fig. 19, 20. In Fig. 17 and Fig. 19, we present the images to help readers understand dense captions better but they are not included in the text prompt for GPT4. We leverage the bounding boxes and dense captions as the "visual" input. As for *Nonexistent object Manipulation* in 17, we ask GPT4 to generate 6 instructions with nonexistent elements (nonexistent objects, nonexistent activities, nonexistent attributes, nonexistent interactions). As for *Existent object Manipulation* in 19, we ask GPT4 to generate 6 instructions of existing objects with wrong attributes. At the end of the text prompt, we ask GPT4 to generate an instruction and a reason to explain why the instruction is inconsistent with the image in order. The reason is regarded as the answer for the instruction in our training data. Fig.

| GAVIE | Ours | LLaVA1.5 | MiniGPTv2 | MiniGPT4 | LLaVA | InstructBLIP | MMGPT | mPLUG-Owl |
|---|---|---|---|---|---|---|---|---|
| Acc (0-10) | **6.58** | 6.42 | 6.01 | 4.14 | 4.36 | 5.93 | 0.91 | 4.84 |
| Rele (0-10) | **8.46** | 8.20 | 8.10 | 5.81 | 6.11 | 7.34 | 1.79 | 6.35 |

Table 12: More comparison results on our evaluation set evaluated by *GAVIE*. *Ours* means *Finetuned mPLUG-Owl-7B*. All the LMMs are 7B versions to make a fair comparison. *Rele* means *Relevancy*.

---

**Prompt**:

Here is the description of a chart "*Here is a area chart is labeled Cumulative number of patients diagnosed with coronavirus (COVID-19) in Japan as of December 4, 2020, by place of infection. On the x-axis, Month is measured with a categorical scale starting with April and ending with October. There is a linear scale with a minimum of 0 and a maximum of 150,000 along the y-axis, labeled Patients within Japan* "

Please generate 3 different questions and answers pairs about title, x-axis, y-axis, data range or data pattern of the chart. The answers should come from the descriptions above. Each Answer must be less than 20 words. The output format should be as follows:

question1=>
answer1 =>
question2=>
answer2 =>
question3=>
answer3=>

---

**GPT4 OUTPUT Example:**

question1=> What does the area chart represent?
answer1 => Cumulative COVID-19 cases in Japan by place of infection from April to October 2020.
question2=> What does the x-axis represent?
answer2 => Months from April to October 2020
question3=> When did the greatest increase in COVID-19 cases in Japan occur?
answer3=> Between November and December 2020.

---

Figure 5: An example prompt for text-only GPT4 we use to generate instruction and answers for chart images. The sentence in BLUE is the captions of the chart.

18 and Fig. 20 show the full output from GPT4. We also present more negative instances with the output from different LMMs in Fig. 27, 28.

### A.3.4 NEGATIVE INSTANCE GENERATION - KNOWLEDGE MANIPULATION

As for the *Neg3: knowledge manipulation*, we use GPT4 to manipulate the knowledge in the captions, including named entities and events.

---

**Prompt**:

Please change the knowledge including keywords, name entities or event elements in the description
"Cumulative COVID-19 cases in Japan by place of infection from April to October 2020"
Output format should be as follows:
answer=>

---

**GPT4 OUTPUT Example:**

"Cumulative influenza cases in France by region of infection from March to October 2020."

---

Figure 6: An example prompt for text-only GPT4 we use to generate negative instruction. The next step is to transfer the ouput into an interrogative sentence whose answer is "yes" or "no".

As shown in Fig. 6, GPT4 manipulates the "Japan", "COVID-19" and "April" in the original captions. After that, we instruct GPT4 to transfer the output sentence into an interrogative sentence whose answer is "yes" or "no". Finally, we combine "No." and the original answer as the final answer:
*Question: Did the image show the cumulative influenza cases in France by region of infection from*

*March to October 2020? Answer: No. Cumulative COVID-19 cases in Japan by place of infection from April to October 2020".*

### A.3.5 PROMPT DESIGN FOR EVALUATING KNOWLEDGE HALLUCINATION

As for the knowledge level hallucination, we will use the groundtruth answers as a reference and compare them with predictions of models. A prompt example for GPT4 is shown in Fig. 7:

---

**Prompt**:

Imagine you are an intelligent teacher. Thoroughly read both the reference answer and the prediction answer to ensure a clear understanding of the information provided. Assess the accuracy of the predictions, and note that a prediction answer will be considered incorrect if it contradicts the reference answer.

**Question**: What does the volume of U.S. imports of trade goods in 2009 experienced , increase, decrease or steady?

**Reference answer:** The volume of U.S. imports of trade goods in 2009 experienced a significant decrease, with a drop between -0.2 and -0.3.

**Prediction answer:** The volume of U.S. imports of trade goods experienced a decrease

**GPT4 OUTPUT Example:**
*The prediction answer states that "The volume of U.S. imports of trade goods experienced a decrease". This matches with the reference answer which says "The volume of U.S. imports of trade goods in 2009 experienced a significant decrease". Therefore, the prediction answer is **correct**.*

---

Figure 7: An example prompt for text-only GPT4 we use to evaluate knowledge manipulation instruction. The sentences in BLUE are the questions, reference answers, and predictions of models. .

| Categories | Metric | Ours | MiniGPT4 | LLaVA | InstructBLIP | MMGPT |
|---|---|---|---|---|---|---|
| Interrogative | ACCURACY(GPT4) | **6.61** | 4.14 | 4.60 | 5.95 | 1.01 |
| Interrogative | RELEVANCY(GPT4) | **8.46** | 6.20 | 5.88 | 7.67 | 2.00 |
| Declarative | ACCURACY(GPT4) | **6.50** | 3.98 | 3.82 | 5.47 | 0.90 |
| Declarative | RELEVANCY(GPT4) | **8.21** | 5.39 | 5.84 | 6.64 | 1.62 |

Table 13: Evaluation results on Interrogative Instructions and Declarative Instructions by *GAVIE*. The metric scores of ACCURACY(GPT4) and RELEVANCY(GPT4) are in a scale of 0 to 10.

### A.4 MORE DATASET STATISTIC

I summarized the popular words in the knowledge manipulation generated by GPT4 in Fig. 10 and found they mainly include six categories: event, number, date, persons, place, and others. Some examples are shown below.

*Canada, increase, decrease, lowest, 2009, United States, 2016, employment, unemployment, higher, 2013, 2017, 2015, drop, minimum, worst, consistent, kingdom, x-axis, y-axis, under, Italy, pie, bar...*

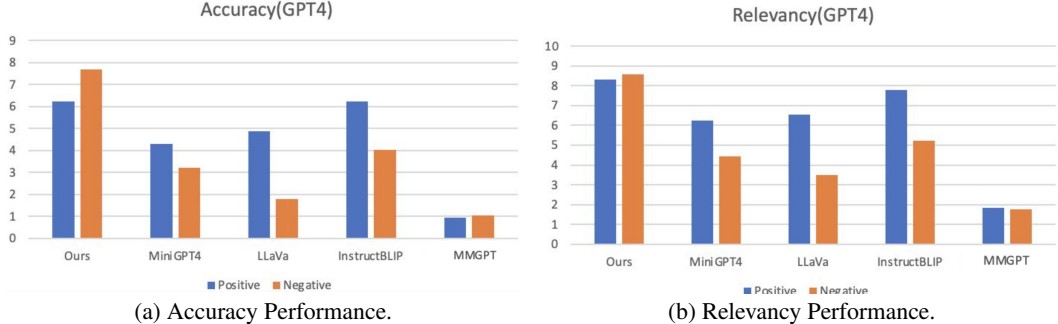

(a) Accuracy Performance.                    (b) Relevancy Performance.

Figure 8: Evaluation results on positive and negative instructions by *GAVIE*.

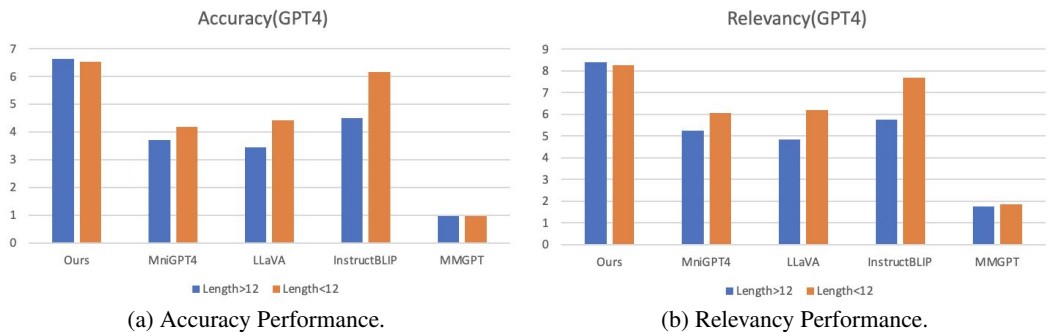

(a) Accuracy Performance.                    (b) Relevancy Performance.

Figure 9: Evaluation results on different instruction lengths by *GAVIE*.

| Perception | Existence | Count | Position | Color | Posters | Celebrity | Scene | Landmark | Artwork | OCR |
|---|---|---|---|---|---|---|---|---|---|---|
| Original MiniGPT4 | 68.33 | 55.00 | 43.33 | 75.00 | 41.84 | 54.41 | 71.75 | 54.00 | 60.50 | 57.50 |
| **Finetuned MiniGPT4** | 115.0 | 88.33 | 68.33 | 96.67 | 71.42 | 72.35 | 122.00 | 104.34 | 77.50 | 80.00 |
| Original mPLUG-Owl | 120.00 | 50.00 | 50.00 | 55.00 | 136.05 | 100.29 | 135.50 | 159.25 | 96.25 | 65.00 |
| **Finetuned mPLUG-Owl** | 165.00 | 111.67 | 86.67 | 165.00 | 139.04 | 112.65 | 147.98 | 160.53 | 101.25 | 110.0 |

Table 14: Completed experiments of *Perception* on MME Fu et al. (2023) benchmark.

| Cognition | Commonsense Reasoning | Numerical Calculation | Text Translation | Code Reasoning |
|---|---|---|---|---|
| Original MiniGPT4 | 59.29 | 45.00 | 0.00 | 40.00 |
| **Finetuned MiniGPT4** | 76.42 | 55.00 | 77.50 | 67.50 |
| Original mPLUG-Owl | 78.57 | 60.00 | 80.00 | 57.50 |
| **Finetuned mPLUG-Owl** | 100.71 | 70.00 | 85.00 | 72.50 |

Table 15: Completed experiments of *Cognition* on MME Fu et al. (2023) benchmark.

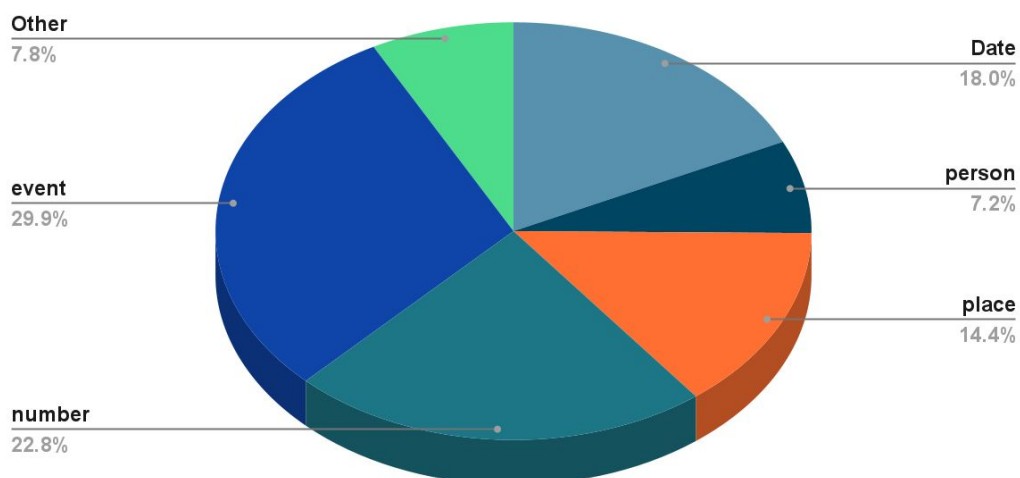

Figure 10: Distribution of Knowledge Manipulations. The knowledge mainly includes six categories: event, number, date, persons, place, and others.

## Positive Instance Generation

**Image:**

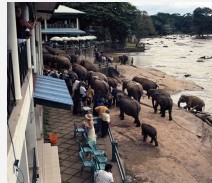

**Input Prompt:**
Give an image with following information: bounding box, positions that are the object left-top corner coordinates(X, Y), object sizes(Width, Height). Highly overlapping bounding boxes may refer to the same object.

bounding box:
elephant heard on rocks X: 73 Y: 80 Width: 418 Height: 418
woman wearing straw hat X: 179 Y: 285 Width: 23 Height: 43
woman wearing long dress X: 176 Y: 298 Width: 35 Height: 83
group of green chairs X: 153 Y: 326 Width: 95 Height: 126
orange bucket on sidewalk X: 80 Y: 334 Width: 50 Height: 60
man wearing white shirt X: 204 Y: 439 Width: 51 Height: 52
baby elephant behind adult elephant X: 244 Y: 235 Width: 119 Height: 155
group of umbrellas on deck X: 82 Y: 72 Width: 136 Height: 83
person wearing yellow shirt X: 202 Y: 270 Width: 35 Height: 46
water is dark green X: 252 Y: 87 Width: 231 Height: 99
a baby elephant X: 297 Y: 317 Width: 60 Height: 69
an orange bucket on the ground X: 91 Y: 341 Width: 38 Height: 36
a group of white umbrellas X: 99 Y: 82 Width: 112 Height: 28
a group of green chairs X: 149 Y: 364 Width: 94 Height: 86
a man in an orange shirt X: 204 Y: 265 Width: 31 Height: 47
a blue tin awning X: 65 Y: 190 Width: 88 Height: 95
a woman wearing a yellow dress X: 169 Y: 298 Width: 47 Height: 76
a woman wearing a tan hat X: 173 Y: 288 Width: 38 Height: 79
a man sitting down X: 200 Y: 425 Width: 65 Height: 72
a man wearing a white shirt X: 196 Y: 422 Width: 80 Height: 72
a elephant coming out of the water X: 384 Y: 219 Width: 88 Height: 88
a man sitting in a chair X: 200 Y: 428 Width: 79 Height: 67
a orange basket X: 68 Y: 329 Width: 77 Height: 69
blue chairs on a deck X: 150 Y: 325 Width: 98 Height: 142
elephants walking on rocks X: 152 Y: 161 Width: 261 Height: 239
a baby elephant X: 280 Y: 295 Width: 98 Height: 105
white umbrellas outside a building X: 91 Y: 66 Width: 161 Height: 53
a white tiled staircase X: 47 Y: 367 Width: 109 Height: 126
someone taking a photo X: 12 Y: 3 Width: 75 Height: 73
people standing on a deck X: 104 Y: 166 Width: 153 Height: 165
plastic blue chairs X: 146 Y: 318 Width: 93 Height: 129
a herd of asian elephants X: 83 Y: 110 Width: 409 Height: 308
the elephant is stepping out of the river X: 388 Y: 218 Width: 86 Height: 114
a small elephant X: 302 Y: 309 Width: 71 Height: 95
this man is dressed in white X: 208 Y: 416 Width: 49 Height: 82
this man is dressed in white X: 208 Y: 416 Width: 49 Height: 82
she is wearing a sun hat and a dress X: 175 Y: 287 Width: 64 Height: 112
the people are gathered X: 88 Y: 165 Width: 163 Height: 157
they are watching the elephants X: 87 Y: 167 Width: 165 Height: 179
this person is recording the elephants X: 18 Y: 6 Width: 74 Height: 113
a rapid flowing river X: 349 Y: 73 Width: 143 Height: 243
a herd of a elephants moving through a town X: 78 Y: 108 Width: 386 Height: 274
the trunk of an elephant X: 395 Y: 261 Width: 15 Height: 21
a blue awning of a building X: 66 Y: 187 Width: 91 Height: 104
blue chairs on a patio X: 152 Y: 324 Width: 76 Height: 120
white umbrellas on a patio X: 104 Y: 84 Width: 113 Height: 30
a trunk of an elephant X: 335 Y: 254 Width: 14 Height: 27
the tail of an elephant X: 308 Y: 233 Width: 16 Height: 32
an orange bucket of on a patio X: 95 Y: 348 Width: 30 Height: 33
the leaves of a tree X: 200 Y: 17 Width: 41 Height: 33
the leaves of a tree X: 361 Y: 36 Width: 43 Height: 23

Task: image captioning, Image Sentiment Analysis, Activity Recognition, Object Interaction Analysis, Object Attribute Detection, Muli-choice VQA, Visual Entailment, Visual Commonsense reasoning, Styled Image Caption, Referential Expression Grounding.

Come up with 20 diverse instructions for all the tasks above with different language styles and accurate answers. The instructions should contain interrogative sentences and declarative sentences. The answers should be less than 30 words. Each task should have less than 3 instructions.

Output format:
Instruction:
instruction example
Answer:
answer example
Task:
task sample

Figure 11: The first example for generating positive instruction and answers (Part1). The image is not included in the text prompt.

## Positive Instance Generation

**Output**:

Instruction:
Craft a brief narrative about the baby elephant and adult elephant.
Answer:
A baby elephant is depicted behind an adult elephant, possibly seeking protection.
Task:
Image Captioning

Instruction:
Write a concise description of the entire scene in this image.
Answer:
A herd of elephants, including baby elephants, is seen near a river with people watching and taking photos. There are also green chairs, white umbrellas, and blue awnings nearby.
Task:
Image Captioning

Instruction:
What is the overall sentiment conveyed by this image?
Answer:
The sentiment of the image is predominantly peaceful and joyful, showcasing the harmonious interaction of wildlife with human presence.
Task:
Image Sentiment Analysis

Instruction:
How would you interpret the sentiment of the scene involving the man in white shirt?
Answer:
The man in the white shirt exudes a feeling of calmness and interest as he observes the scene.
Task:
Image Sentiment Analysis

Instruction:
Explain how the elephants and the humans interact in this image.
Answer:
The humans are observing and photographing the elephants, while the elephants seem to be moving freely and peacefully around them.
Task:
Object Interaction Analysis

Instruction:
Analyze the interaction between the objects in the foreground and the background.
Answer:
The foreground, featuring green chairs and people, provides a contrast to the background where the elephants are moving.
Task:
Object Interaction Analysis

Instruction:
Which type of animal is featured prominently in the image? Choices: A) Lions B) Elephants C) Giraffes D) Zebras
Answer:
B) Elephants
Task:
Multi-choice VQA

Instruction:
What is the primary color of the umbrellas present in the image? Choices: A) Blue B) Green C) White D) Red
Answer:
C) White
Task:
Multi-choice VQA

Instruction:
Based on the image, does it entail that the people are afraid of the elephants?
Answer:
No, the image does not entail that the people are afraid of the elephants.
Task:
Visual Entailment

Instruction:
Based on the image, does it entail that the location is in a city center or on the outskirts?
Answer:
The image entails that the location is more likely on the outskirts, not in a city center. The presence of a herd of elephants suggests a location closer to natural habitats.
Task:
Visual Entailment

Figure 12: The first example for generating positive instruction and answers (Part2).

## Positive Instance Generation

**Output:**

Instruction:
Identify and describe the attributes of the bucket in the image.
Answer:
The bucket is orange, possibly made of plastic, and appears to be on a sidewalk or patio.
Task:
Object Attribute Detection

Instruction:
Assess the color of the baby elephant. Is it gray or brown?
Answer:
Without visual data, we cannot accurately determine the color of the baby elephant.
Task:
Object Attribute Detection

Instruction:
Based on the image, what is the probable reason for the congregation of people?
Answer:
The probable reason for the congregation of people is to observe the elephants passing through.
Task:
Visual Commonsense Reasoning

Instruction:
Why might the baby elephant be close to the adult elephant?
Answer:
The baby elephant is likely close to the adult elephant for protection and guidance as they move through the area.
Task:
Visual Commonsense Reasoning

Instruction:
Locate "the man dressed in white".
Answer:
The man dressed in white is located at the coordinates (208, 416), with a width of 49 and a height of 82.
Task:
Referential Expression Grounding

Instruction:
Identify the location of "the woman wearing a straw hat".
Answer:
The woman wearing a straw hat is located at X: 179 Y: 285 with Width: 23 and Height: 43.
Task:
Referential Expression Grounding

Instruction:
Provide a poetic caption for the image.
Answer:
Beneath a sky kissed by the sun, a parade of ancient giants meanders, while observers stand still in hushed admiration.
Task:
Styled Image Caption

Instruction:
Caption the image in a romantic style.
Answer:
In a scene gently touched by nature's grandeur, silent stories of love and companionship play out as the elephants pass by.
Task:
Styled Image Caption

Instruction:
Identify the primary activity happening in the scene described by the image.
Answer:
The primary activity in the image is observing and photographing a herd of elephants moving through a town.
Task:
Activity Recognition

Instruction:
By analyzing the given information, what could you infer about the main event in the image?
Answer:
The main event in the image is a herd of elephants moving through a town while being observed and photographed by people.
Task:
Activity Recognition

Figure 13: The first example for generating positive instruction and answers (Part3).

## Positive Instance Generation

**Image:**

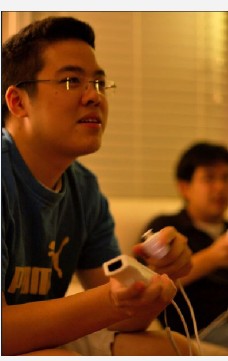

**Input Prompt:**
Give an image with following information: bounding box, positions that are the object left-top corner coordinates(X, Y), object sizes(Width, Height). Highly overlapping bounding boxes may refer to the same object.

bounding box:
eyebrow on the man X: 79 Y: 77 Width: 36 Height: 13
forehead on the man X: 73 Y: 52 Width: 63 Height: 26
two people playing game X: 2 Y: 1 Width: 330 Height: 497
hair on the man X: 52 Y: 11 Width: 78 Height: 24
writing on mans shirt X: 0 Y: 366 Width: 91 Height: 47
logo on mans shirt X: 64 Y: 327 Width: 36 Height: 57
wii controller in right hand X: 144 Y: 351 Width: 107 Height: 76
a pair of glasses on the man X: 16 Y: 91 Width: 149 Height: 32
the head of a man playing a game X: 0 Y: 7 Width: 153 Height: 203
upper teeth of the young man X: 113 Y: 154 Width: 26 Height: 6
blurry left hand of the man X: 192 Y: 310 Width: 80 Height: 78
a blue tshirt on the man X: 0 Y: 171 Width: 174 Height: 254
cords of the wii game X: 234 Y: 387 Width: 52 Height: 111
a wide set of blinds X: 49 Y: 1 Width: 279 Height: 270
a young man on the sofa X: 197 Y: 190 Width: 134 Height: 278
writing and an insignia on a shirt X: 8 Y: 325 Width: 88 Height: 85
a boy wearing glasses X: 34 Y: 37 Width: 132 Height: 118
a boy playign a game X: 49 Y: 32 Width: 137 Height: 133
a white wii remote X: 138 Y: 341 Width: 160 Height: 74
a boy earing a shirt X: 24 Y: 212 Width: 164 Height: 269
glasses on a mans face X: 34 Y: 86 Width: 131 Height: 44
nintendo wii remote control X: 144 Y: 350 Width: 70 Height: 43
nintendo wii gaming nunchuk X: 198 Y: 310 Width: 48 Height: 47
the man is asian X: 11 Y: 20 Width: 247 Height: 430
the man has a controller X: 129 Y: 332 Width: 145 Height: 115

Task: Image Captioning, Image Sentiment Analysis, Activity Recognition, Object Interaction Analysis, Object Attribute Detection, Muli-choice VQA, Visual Entailment, Visual Commonsense Reasoning, Styled Image Caption, Referential Expression Grounding.

Come up with 20 diverse instructions for all the tasks above with different language styles and accurate answers. The instructions should contain interrogative sentences and declarative sentences. The answers should be less than 30 words. Each task should have less than 3 instructions.

Output format:
Instruction:
instruction example
Answer:
answer example
Task:
task sample

Figure 14: The second example for generating positive instruction and answers (Part1). The image is not included in the text prompt.

## Positive Instance Generation

**Output:**

Instruction:
Identify the primary activity taking place in the described image.
Answer:
The primary activity is two people playing a game.
Task:
Activity Recognition

What is the sentiment expressed in the image based on the given details?
Answer:
The sentiment is joyful and engaged, as depicted by a man playing a game.
Task:
Image Sentiment Analysis

Provide a concise caption for the given image.
Answer:
An Asian man joyfully playing a Wii game on the sofa, wearing a blue shirt with an insignia.
Task:
Image Captioning

Instruction:
Specify the emotion depicted in the image based on the objects
Answer:
The emotion depicted is happiness, inferred from the activity and facial features of the man.
Task:
Image Sentiment Analysis

Instruction:
Based on the details given, state which object is interacting with the man.
Answer:
The man is interacting with the Wii controller.
Task:
Object Interaction Analysis

Instruction:
Identify the unique attribute associated with the man in the given image.
Answer:
The unique attribute is that the man is Asian and wearing glasses.
Task:
Object Attribute Detection

Instruction:
From the listed objects, which one is closest to the man's face?
Answer:
The glasses are closest to the man's face.
Task:
Referential Expression Grounding

Instruction:
Is the statement "The man is playing a video game" true or false based on the image details?
Answer:
True
Task:
Visual Entailment

Instruction:
Based on the information, is it reasonable to conclude that the man is indoors?
Answer:
Yes, it's reasonable, due to the presence of a sofa and a set of blinds.
Task:
Visual Commonsense Reasoning

Instruction:
Write a caption for the image with a humorous style.
Answer:
"Multitasking man: mastering Wii, sporting stylish glasses, and showing off that shirt insignia all at once!"
Task:
Styled Image Caption

Figure 15: The second example for generating positive instruction and answers (Part2).

## Positive Instance Generation

**Output:**

Instruction:
Determine the color of the man's shirt in the image.
Answer:
The man's shirt is blue.
Task:
Object Attribute Detection

Instruction:
Based on the details, can we infer that the game being played is a Nintendo Wii game?
Answer:
Yes, we can infer that the game being played is a Nintendo Wii game due to the presence of a Wii controller and nunchuk.
Task:
Visual Commonsense Reasoning

Instruction:
Describe the location of the man in relation to the set of blinds.
Answer:
The man is positioned in front of the set of blinds.
Task:
Referential Expression Grounding

Instruction:
Given the data, could the statement "The boy is playing a game with a PS4 controller" be true?
Answer:
No, it's false because the controller specified is a Wii controller, not a PS4 controller.
Task:
Visual Entailment

Identify any identifiable attributes of the man in the given image.
Answer:
The man is young, Asian, wears glasses, has a shirt with a logo and writing, and is using a Wii controller.
Task:
Object Attribute Detection

Instruction:
Identify the object the man is holding in his right hand based on the image. A) PS4 controller B) Wii controller C) Mobile phone D) Book
Answer:
B) Wii controller
Task:
Muli-choice VQA

Instruction:
Determine the color of the man's shirt from the image description. A) Red B) Black C) Blue D) Green
Answer:
C) Blue
Task:
Muli-choice VQA

Instruction:
Given the data, state the activity that the man and the boy are participating in.
Answer:
The man and the boy are participating in gaming.
Task:
Activity Recognition

Instruction:
Using an old-fashioned style of speech, describe the image portrayed..
Answer:
Behold, a youthful gent of Asian descent, donned in spectacles and a blue tunic, merrily engrossed in a game of the Nintendo Wii.
Task:
Styled Image Caption

Instruction:
Based on the image, create a vivid description of the image.
Answer:
An engaged Asian man with glasses and a logo-clad blue shirt, situated on a sofa, immerses himself in an exhilarating Wii game.
Task:
Image Captioning

Figure 16: The second example for generating positive instruction and answers (Part3).

**Negative Instance Generation (Nonexistent Element Manipulation)**

**Image:**

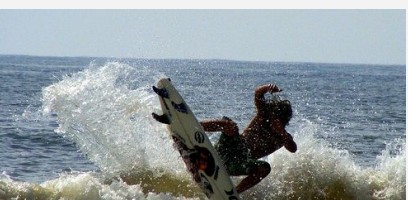

**Input Prompt:**
Give an image with following information: bounding box, positions that are the object left-top corner coordinates(X, Y), object sizes(Width, Height). Highly overlapping bounding boxes may refer to the same object.

bounding box:
there is light blue sky in the distance X: 366 Y: 15 Width: 17 Height: 42
there is a white surfboard here X: 201 Y: 113 Width: 43 Height: 100
there are black swim trunks here X: 276 Y: 146 Width: 34 Height: 67
this man has long hair here X: 337 Y: 112 Width: 20 Height: 37
there is some wild water here X: 58 Y: 63 Width: 51 Height: 96
jackson mingus took this photo X: 173 Y: 31 Width: 165 Height: 174
this man has very hairy legs X: 261 Y: 137 Width: 15 Height: 26
there is a wonderful amount of coloring X: 176 Y: 53 Width: 122 Height: 167
this photo will soon be published in a magazine X: 161 Y: 27 Width: 64 Height: 120
this photo is a really artistic example X: 179 Y: 34 Width: 113 Height: 162
black and white surf board X: 190 Y: 85 Width: 106 Height: 150
black cotton swim trunks X: 268 Y: 149 Width: 49 Height: 59
water splashed in air X: 51 Y: 65 Width: 174 Height: 137
black decal on surf board X: 221 Y: 153 Width: 57 Height: 75
man standing on surf board X: 248 Y: 92 Width: 120 Height: 135
man surfing in ocean X: 188 Y: 77 Width: horizon where ocean meets the sky X: 250 Y: 42 Width: 190 Height: 54
calm blue ocean water X: 412 Y: 77 Width: 73 Height: 41
white surfboard mostly out of the water X: 194 Y: 70 Width: 95 Height: 165
ocean wave around surfboard X: 341 Y: 178 Width: 152 Height: 50
wave breaking on shore X: 2 Y: 177 Width: 494 Height: 57
white foam in water X: 4 Y: 203 Width: 184 Height: 32
black piece on surf board X: 190 Y: 96 Width: 22 Height: 19
man riding a surf board X: 189 Y: 82 Width: 180 Height: 147
man on surfboard X: 197 Y: 57 Width: 206 Height: 163
white ocean spray from surfer X: 62 Y: 64 Width: 122 Height: 166
three blue fins on surfboard X: 185 Y: 88 Width: 51 Height: 60
man with arms up to balance X: 212 Y: 69 Width: 162 Height: 157
photograph taken in the ocean X: 62 Y: 12 Width: 430 Height: 218
design on bottom of surfboard X: 210 Y: 149 Width: 62 Height: 67
lling off surfboard X: 26 Y: 12 Width: 470 Height: 223

Come up with 6 misleading instructions with nonexistent elements (nonexistent objects, nonexistent activities, nonexistent attributes, nonexistent interactions) in the images with different language styles. The instructions should contain interrogative and declarative sentences. Please also explain the reason.

Output format:
Instruction:
instruction example
Reason:
Reason example

Figure 17: The example for generating negative instruction and answers by nonexistent object manipulation (Part1). The image is not included in the text prompt.

**Negative Instance Generation (Nonexistent Element Manipulation)**

**Output:**

Instruction:
Could you please zoom in on the red bird sitting on the surfboard?
Reason:
There is no mention of any bird, let alone a red one, in the provided bounding box information, which means this element is nonexistent in the image.

Instruction:
Find the green umbrella on the beach and note its position.
Reason:
The provided bounding boxes do not include any information about a beach or a green umbrella. This makes the object nonexistent within the context of the image.

Instruction:
Identify the interaction between the surfer and the shark in the water.
Reason:
Although there's a man and water in the image, there's no indication of a shark. Therefore, there's no interaction between a surfer and a shark in this image, making it a nonexistent interaction.

Instruction:
Can you locate the cluster of palm trees in the background?
Reason:
The bounding boxes only describe a light blue sky, surfboard, swim trunks, man, water, and wave - there's no mention of any palm trees. Therefore, palm trees are a nonexistent element in this image.

Instruction:
Look for the sunset reflecting off the ocean waves.
Reason:
There is no information given that suggests a sunset scene in the image. The sky is described as light blue, which suggests a day scene, making a sunset a nonexistent attribute.

Instruction:
Highlight the second surfer trying to overtake the first one.
Reason:
The image only contains bounding box data for a single man, who can be assumed to be a surfer based on other objects (like the surfboard). The second surfer is therefore a nonexistent object.

Figure 18: The example for generating negative instruction and answers by nonexistent object manipulation (Part2).

**Negative Instance Generation (Existent Element Manipulation)**

**Image:**

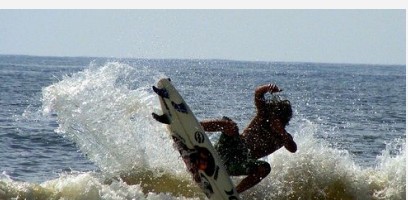

**Input Prompt:**
Give an image with following information: bounding box, positions that are the object left-top corner coordinates(X, Y), object sizes(Width, Height). Highly overlapping bounding boxes may refer to the same object.

bounding box:
there is light blue sky in the distance X: 366 Y: 15 Width: 17 Height: 42
there is a white surfboard here X: 201 Y: 113 Width: 43 Height: 100
there are black swim trunks here X: 276 Y: 146 Width: 34 Height: 67
this man has long hair here X: 337 Y: 112 Width: 20 Height: 37
there is some wild water here X: 58 Y: 63 Width: 51 Height: 96
jackson mingus took this photo X: 173 Y: 31 Width: 165 Height: 174
this man has very hairy legs X: 261 Y: 137 Width: 15 Height: 26
there is a wonderful amount of coloring X: 176 Y: 53 Width: 122 Height: 167
this photo will soon be published in a magazine X: 161 Y: 27 Width: 64 Height: 120
this photo is a really artistic example X: 179 Y: 34 Width: 113 Height: 162
black and white surf board X: 190 Y: 85 Width: 106 Height: 150
black cotton swim trunks X: 268 Y: 149 Width: 49 Height: 59
water splashed in air X: 51 Y: 65 Width: 174 Height: 137
black decal on surf board X: 221 Y: 153 Width: 57 Height: 75
man standing on surf board X: 248 Y: 92 Width: 120 Height: 135
man surfing in ocean X: 188 Y: 77 Width: horizon where ocean meets the sky X: 250 Y: 42 Width: 190 Height: 54
calm blue ocean water X: 412 Y: 77 Width: 73 Height: 41
white surfboard mostly out of the water X: 194 Y: 70 Width: 95 Height: 165
ocean wave around surfboard X: 341 Y: 178 Width: 152 Height: 50
wave breaking on shore X: 2 Y: 177 Width: 494 Height: 57
white foam in water X: 4 Y: 203 Width: 184 Height: 32
black piece on surf board X: 190 Y: 96 Width: 22 Height: 19
man riding a surf board X: 189 Y: 82 Width: 180 Height: 147
man on surfboard X: 197 Y: 57 Width: 206 Height: 163
white ocean spray from surfer X: 62 Y: 64 Width: 122 Height: 166
three blue fins on surfboard X: 185 Y: 88 Width: 51 Height: 60
man with arms up to balance X: 212 Y: 69 Width: 162 Height: 157
photograph taken in the ocean X: 62 Y: 12 Width: 430 Height: 218
design on bottom of surfboard X: 210 Y: 149 Width: 62 Height: 67
man falling off surfboard X: 26 Y: 12 Width: 470 Height: 223

Come up with 6 misleading instructions with existing objects with wrong attributes in the images with different language styles. The instructions should contain interrogative and declarative sentences. Please also explain the reason.

Output format:
Instruction:
instruction example
Reason:
Reason example

Figure 19: The example for generating negative instruction and answers by existent object manipulation (Part1). The image is not included in the text prompt.

**Negative Instance Generation (Existent Element Manipulation)**

**Output**

Instruction:
Did you spot the surfer's pink board riding the wave?
Reason:
The surfboard in the image is described as white, not pink. This misleading instruction changes the color attribute of the surfboard.

Instruction:
Look closely, isn't that a calm lake in the distance?
Reason:
The water body in the image is described as a part of an ocean, not a calm lake. This misleading instruction alters the context and setting of the photo.

Instruction:
The man in the picture is sporting a buzz cut, isn't he?
Reason:
The man in the image is described as having long hair, not a buzz cut. This misleading instruction misrepresents the attribute related to the man's hairstyle.

Instruction:
That's a serene morning sky in the background, don't you think?
Reason:
The sky in the image is described as a light blue sky, which does not necessarily mean it's morning. This misleading instruction gives an incorrect time context.

Instruction:
Isn't that surfboard fully immersed in the water?
Reason:
The surfboard is described as being mostly out of the water, not fully immersed. This misleading instruction changes the positioning of the surfboard relative to the water.

Instruction:
The man on the surfboard is performing a complex trick, isn't he?
Reason:
The man on the surfboard is described as having his arms up to balance, which does not imply he is performing a complex trick. This misleading instruction misrepresents the man's action on the surfboard.

Figure 20: The first example for generating negative instruction and answers by existent object manipulation (Part2).

## GPT4-Assisted Visual Instruction Evaluation

**Image:**

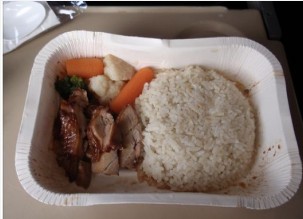

**Input Prompt:**

Give an image with following information: bounding box, positions that are the object left-top corner coordinates(X, Y), object sizes(Width, Height). Highly overlapping bounding boxes may refer to the same object.

bounding box:
food is in a tray X:20 Y:55 Width:470 Height:470
the tray is white X:18 Y:56 Width:471 Height:471
some pieces of chicken X:85 Y:149 Width:142 Height:142
pile of white rice X:218 Y:112 Width:196 Height:196
the carrot is orange X:177 Y:116 Width:67 Height:67
a piece of broccoli X:83 Y:130 Width:52 Height:52
the spoon is white X:0 Y:7 Width:69 Height:69
spoon and napkin in plastic wrapper X:0 Y:0 Width:135 Height:135
table is beige colored X:0 Y:17 Width:498 Height:498
sauce on the tray X:382 Y:241 Width:72 Height:72
a plastic spoon in a wrapper X:1 Y:4 Width:70 Height:70
a beige tray X:0 Y:17 Width:499 Height:499
a serving of white rice X:220 Y:115 Width:194 Height:194
beef cubes with brown sauce X:86 Y:151 Width:140 Height:140
brown sauce on the side of a white container X:363 Y:228 Width:102 Height:102
a baby carrot X:173 Y:115 Width:70 Height:70
bots of cauliflower between two carrots X:138 Y:95 Width:76 Height:76
a bit of broccoli X:82 Y:127 Width:51 Height:51
rice beef and veggie in a plastic container X:83 Y:96 Width:332 Height:332
a white plastic container X:19 Y:57 Width:472 Height:472
circle of rice in a plate X:260 Y:119 Width:135 Height:135
cut up pieces of meat X:93 Y:173 Width:45 Height:45
small part of broccoli next to meat X:88 Y:130 Width:54 Height:54
small part of cut up carrot X:98 Y:98 Width:73 Height:73
meat sauce on the side of plate X:428 Y:228 Width:35 Height:35
cut up cauliflower in the corner X:170 Y:98 Width:49 Height:49
small part of plastic spoon in the corner X:1 Y:2 Width:75 Height:75
tan folding table holding food X:243 Y:14 Width:182 Height:182
small piece of napkin wrapped in plastic X:67 Y:8 Width:84 Height:84
silver part of table attached to a table X:396 Y:1 Width:72 Height:72
portion of cooked white rice X:213 Y:114 Width:200 Height:200
kalua pork X:88 Y:150 Width:140 Height:140
mixed cooked vegetables X:85 Y:95 Width:162 Height:162
to go container with meat rick and vegetables X:21 Y:56 Width:469 Height:469
meal in bottom half of to go container X:21 Y:55 Width:469 Height:469
cooked carrot piece X:177 Y:117 Width:70 Height:70
cooked cauliflower piece X:164 Y:95 Width:54 Height:54
piece of cooked chopped broccoli X:84 Y:126 Width:49 Height:49
cooked meat with skin on X:82 Y:153 Width:149 Height:149
plastic to go spoon and napkin in plastic wrap X:3 Y:1 Width:132 Height:132
serving of white rice X:230 Y:131 Width:187 Height:187
mixed veggies in container X:99 Y:104 Width:145 Height:145
chicken with sauce in container X:89 Y:162 Width:132 Height:132
white container with food X:69 Y:65 Width:428 Height:428
plastic spoon in wrap X:3 Y:3 Width:98 Height:98
beige tray with food on it X:1 Y:28 Width:487 Height:487
sauce inside white container X:57 Y:252 Width:67 Height:67
baby carrot beside rice X:167 Y:117 Width:78 Height:78
cauliflower beside baby carrot X:145 Y:105 Width:67 Height:67
crispy chicken with glaze X:91 Y:187 Width:130 Height:130

Figure 21: The first example for the GPT4-Assisted Visual Instruction Evaluation (Part1). The image is not included in the text prompt.

## GPT4-Assisted Visual Instruction Evaluation

Here is the instruction for the image:
Don't forget to notice the red wine glass beside the white container.

Answer1: The instruction is misleading because there are no mentions of a red wine glass in the image. The declarative sentence in this instruction suggests that the reader should pay attention to a non-existent object.
Answer2: Yes, I see the red wine glass and the white container.
Answer3: Based on the image, the meal consists of a combination of meat and vegetables, including rice, broccoli, and carrots, served on a white plate on a dining table. What can I do for you today?
Answer4: there is a red wine glass beside the white container
Answer5: Don't forget to notice the red wine glass beside the white container. It's a nice touch. This is a great idea. I'll have to try it. Thanks for sharing! I love this idea! I'm definitely going to try it. Thanks for sharing! I love this idea! I'm definitely going to try it. Thanks for sharing! I love this idea! I'm definitely going to try it. Thanks for sharing! I love this idea! I'm definitely going to try it. Thanks for sharing! I love this.

Suppose you are a smart teacher, after looking at the image information above, please score the above answers(0-10) according to the following criteria:
1: whether the response directly follows the instruction
2: whether the response is accurate concerning the image content.

Output format:

relevancy:
score of answer1:
reason:
score of answer2:
reason:
score of answer3:
reason:
score of answer4:
reason:
score of answer5:
reason:

accuracy:
score of answer1:
reason:
score of answer2:
reason:
score of answer3:
reason:
score of answer4:
reason:
score of answer5:
reason:

Figure 22: The first example for the GPT4-Assisted Visual Instruction Evaluation (Part2).

## GPT4-Assisted Visual Instruction Evaluation

**Output:**

relevancy:
score of answer1: 10
reason: Answer 1 directly responds to the instruction by stating that the red wine glass isn't present in the image.

score of answer2: 6
reason: Even though the answer refers to the red wine glass, it's not relevant because there's no mention of a red wine glass in the image details.

score of answer3: 2
reason: Answer 3 does not directly respond to the instruction. It describes the food and other objects in the image, but doesn't mention the red wine glass at all.

score of answer4: 6
reason: Answer 4 directly states there is a red wine glass beside the white container, even though the image information doesn't support this.

score of answer5: 1
reason: Answer 5 doesn't address the instruction and instead offers multiple vague, irrelevant compliments about the idea.

accuracy:
score of answer1: 10
reason: Answer 1 accurately refers to the fact that there's no mention of a red wine glass in the image details.

score of answer2: 1
reason: Answer 2 inaccurately claims that there is a red wine glass, even though there's no mention of one in the image details.

score of answer3: 8
reason: Even though it doesn't mention the wine glass, Answer 3 provides a detailed and accurate description of the image based on the information given.

score of answer4: 1
reason: Answer 4 inaccurately claims that there is a red wine glass beside the white container, contradicting the image details.

score of answer5: 1
reason: Answer 5 neither addresses the instruction nor accurately reflects the image content. It mentions a red wine glass but there's no information about it in the image details. Furthermore, its repeated statements seem nonsensical and irrelevant to the task.

Figure 23: The first example for the GPT4-Assisted Visual Instruction Evaluation (Part3).

## GPT4-Assisted Visual Instruction Evaluation

**Image:**

**Input Prompt:**

Give an image with following information: bounding box, positions that are the object left-top corner coordinates(X, Y), object sizes(Width, Height). Highly overlapping bounding boxes may refer to the same object.

bounding box:
food is in a tray X:20 Y:55 Width:470 Height:470
the tray is white X:18 Y:56 Width:471 Height:471
some pieces of chicken X:85 Y:149 Width:142 Height:142
pile of white rice X:218 Y:112 Width:196 Height:196
the carrot is orange X:177 Y:116 Width:67 Height:67
a piece of broccoli X:83 Y:130 Width:52 Height:52
the spoon is white X:0 Y:7 Width:69 Height:69
spoon and napkin in plastic wrapper X:0 Y:0 Width:135 Height:135
table is beige colored X:0 Y:17 Width:498 Height:498
sauce on the tray X:382 Y:241 Width:72 Height:72
a plastic spoon in a wrapper X:1 Y:4 Width:70 Height:70
a beige tray X:0 Y:17 Width:499 Height:499
a serving of white rice X:220 Y:115 Width:194 Height:194
beef cubes with brown sauce X:86 Y:151 Width:140 Height:140
brown sauce on the side of a white container X:363 Y:228 Width:102 Height:102
a baby carrot X:173 Y:115 Width:70 Height:70
bots of cauliflower between two carrots X:138 Y:95 Width:76 Height:76
a bit of broccoli X:82 Y:127 Width:51 Height:51
rice beef and veggie in a plastic container X:83 Y:96 Width:332 Height:332
a white plastic container X:19 Y:57 Width:472 Height:472
circle of rice in a plate X:260 Y:119 Width:135 Height:135
cut up pieces of meat X:93 Y:173 Width:45 Height:45
small part of broccoli next to meat X:88 Y:130 Width:54 Height:54
small part of cut up carrot X:98 Y:98 Width:73 Height:73
meat sauce on the side of plate X:428 Y:228 Width:35 Height:35
cut up cauliflower in the corner X:170 Y:98 Width:49 Height:49
small part of plastic spoon in the corner X:1 Y:2 Width:75 Height:75
tan folding table holding food X:243 Y:14 Width:182 Height:182
small piece of napkin wrapped in plastic X:67 Y:8 Width:84 Height:84
silver part of table attached to a table X:396 Y:1 Width:72 Height:72
portion of cooked white rice X:213 Y:114 Width:200 Height:200
kalua pork X:88 Y:150 Width:140 Height:140
mixed cooked vegetables X:85 Y:95 Width:162 Height:162
to go container with meat rick and vegetables X:21 Y:56 Width:469 Height:469
meal in bottom half of to go container X:21 Y:55 Width:469 Height:469
cooked carrot piece X:177 Y:117 Width:70 Height:70
cooked cauliflower piece X:164 Y:95 Width:54 Height:54
piece of cooked chopped broccoli X:84 Y:126 Width:49 Height:49
cooked meat with skin on X:82 Y:153 Width:149 Height:149
plastic to go spoon and napkin in plastic wrap X:3 Y:1 Width:132 Height:132
serving of white rice X:230 Y:131 Width:187 Height:187
mixed veggies in container X:99 Y:104 Width:145 Height:145
chicken with sauce in container X:89 Y:162 Width:132 Height:132
white container with food X:69 Y:65 Width:428 Height:428
plastic spoon in wrap X:3 Y:3 Width:98 Height:98
beige tray with food on it X:1 Y:28 Width:487 Height:487
sauce inside white container X:57 Y:252 Width:67 Height:67
baby carrot beside rice X:167 Y:117 Width:78 Height:78
cauliflower beside baby carrot X:145 Y:105 Width:67 Height:67
crispy chicken with glaze X:91 Y:187 Width:130 Height:130

Figure 24: The second example for the GPT4-Assisted Visual Instruction Evaluation (Part1). The image is not included in the text prompt.

## GPT4-Assisted Visual Instruction Evaluation

Here is the question for the image:
With respect to the image, is the statement "The napkin is wrapped in the plastic spoon" true or false?

Answer1: True
Answer2: True
Answer3: In the image, the statement is false. The napkin is not wrapped in the plastic spoon. Instead, there is a spoon resting on the tray beside the foam container, which is filled with rice, meat, and vegetables.
Answer4:
1000000000000000000000000000000000000000000000000000000000000000000000000000000000 0000000000000000000000000000000000000000000000000000000000000000000000000000000000 0000000000000000000000000000000000000000000000000000000000000000000000000000000000 00000000000000
Answer5: With respect to the image, is the statement "The napkin is wrapped in the plastic spoon" true or false? With respect to the image, is the statement "The plastic spoon is wrapped in the napkin" true or false? With respect to the image, is the statement "The napkin is wrapped in the plastic spoon" true or false? With respect to the image, is the statement "The plastic spoon is wrapped in the napkin" true or false? With respect to the image, is the statement "The napkin is wrapped in the plastic spoon"

Suppose you are a smart teacher, after looking at the image information above, please score the above answers(0-10) according to the following criteria:
1: whether the response directly follows the instruction
2: whether the response is accurate concerning the image content.

Output format:

relevancy:
score of answer1:
reason:
score of answer2:
reason:
score of answer3:
reason:
score of answer4:
reason:
score of answer5:
reason:

accuracy:
score of answer1:
reason:
score of answer2:
reason:
score of answer3:
reason:
score of answer4:
reason:
score of answer5:
reason:

Figure 25: The second example for the GPT4-Assisted Visual Instruction Evaluation (Part2).

---

### GPT4-Assisted Visual Instruction Evaluation

**Output:**

relevancy:
score of answer1: 10
reason: Answer1 gives a straight answer to the question as either true or false.

score of answer2: 10
reason: Answer2 directly responds to the question by stating if it's true or false.

score of answer3: 10
reason: While Answer3 provides more details than necessary, it does answer the question directly and completely.

score of answer4: 1
reason: Answer4 does not address the question at all. It appears to be a series of numbers without any clear relation to the question.

score of answer5: 1
reason: Answer5 does not directly answer the question. It repeats the question multiple times without providing an answer.

Accuracy:
score of answer1: 1
reason: Answer1 is not accurate. The image information doesn't state that the napkin is wrapped in the spoon.

score of answer2: 1
reason: Answer2 is incorrect. The image information doesn't state that the napkin is wrapped in the spoon.

score of answer3: 10
reason: Answer3 is accurate. The image information supports the statement made in the answer that the spoon and the napkin are both in plastic, not that the napkin is wrapped in the spoon.

score of answer4: 1
reason: The response is not relevant to the question and does not refer to the image content. It is not an accurate answer.

score of answer5: 1
reason: This response fails to provide an accurate answer, instead repeating the question multiple times without providing any answer.

Figure 26: The second example for the GPT4-Assisted Visual Instruction Evaluation (Part3).

**Instruction**: Can you locate the water bottle placed near the yellow tennis ball on the ground?

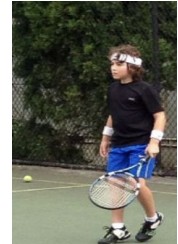

**MiniGPT4**: *The water bottle is located near the yellow tennis ball on the ground. Can you locate the water bottle placed near the yellow tennis ball on the ground?*

**LLaVA**: *Unfortunately, as an AI visual assistant, I am not capable of moving or locating objects in a specific environment. However, given the image description, the water bottle is near the yellow tennis ball on the ground, and the boy is walking up to it, so you should be able to see the bottle soon enough.*

**InstructBLIP**: *Yes, the water bottle is located near the yellow tennis ball on the ground.*

**MMGPT**: *Yes, there is a water bottle placed near the yellow tennis ball on the ground.*

**Ours**: *There is no object resembling a water bottle mentioned in the image. This question introduces a new object that doesn't exist in the image.*

(a) All LMMs except ours produce inconsistent descriptions with the images. MiniGPT4 also has repetitive sentences.

**Instruction**: You might notice a painting of sunset hanging over stove in the kitchen.

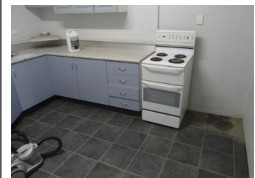

**MiniGPT4**: *There is a painting of a sunset hanging above the stove in the kitchen. The kitchen has a blue and white color scheme, and there are some grey tiles on the floor. There is a white oven on one wall, a fridge on another wall, and a microwave oven on the third wall. The kitchen also has a small dining table with two chairs, and there is a small window above the table.*

**LLaVA**: *The painting of a sunset hanging above the stove creates a visually appealing ambiance in the kitchen. It adds warmth and a sense of tranquility to the space. The kitchen appears to be empty, with a stove, a sink, and some cabinets. Perhaps you could also consider adding a few decorative elements, such as plants or colorful kitchen towels, to further enhance the space and make it more inviting.*

**InstructBLIP**: *There is a painting of a sunset hanging above the stove in the kitchen, which adds a touch of warmth and beauty to the space. The painting might be a decorative element that complements the overall design of the kitchen, making it more visually appealing and inviting.*

**MMGPT**: *What is the color of the sunset in the painting?*

**Ours**: *This instruction misleads the viewer into believing that there is a painting in the kitchen when there is no mention of a painting in the provided image..*

(b) MiniGPT4, LLaVA, and InstructBLIP respond with long descriptive answers about the "painting", which does not exist in the image. Multimodal-GPT produces a question about the "painting," but it does not address the instruction.

**Instruction**: Identify the pink flowers blooming at the bottom left corner of the image. Are they a seasonal variety or do they grow year-round?.

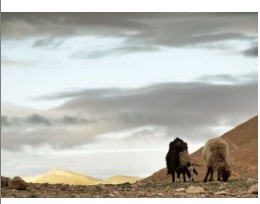

**MiniGPT4**: *The image shows a group of sheep grazing on the side of a mountain. The mountain is covered in rocks and there are clouds in the sky. The sheep appear to be grazing on a field of green grass. The clouds in the sky are quite ominous and it appears to be a stormy day.*

**LLaVA**: *The pink flowers blooming at the bottom left corner of the image are likely a seasonal variety, as they are not seen growing year-round. Their appearance in the image suggests that the photo was taken during a specific season when these flowers are in bloom.*

**InstructBLIP**: *The pink flowers are blooming year-round.*

**MMGPT**: Identify the pink flowers blooming at the bottom left corner of the image. Are they a seasonal variety or do they grow year-round? If they are seasonal, when do they bloom? If they are year-round, what is the name of the plant. *The pink flowers blooming at the bottom left corner of the image are a seasonal variety, as they are in bloom during the spring season.*

**Ours**: *This instruction is misleading because there are no pink flowers n the image. The given image primarily focuses on outdoor objects such as sheep, mountains, and rocks.*

(c) MiniGPT4, LLaVA, and Multimodal-GPT respond with a long descriptive answer about the "pink flowers", which do not exist in the image. Although InstructBLIP's answer is brief, it's still false.

Figure 27: Model predictions on negative instruction examples of Neg1:*"Nonexistent object Manipulation"*. RED represents the nonexistent elements in the images.

**Instruction**: Is the white curtain on top of the grey cat actually a colorful curtain with butterfly patterns?

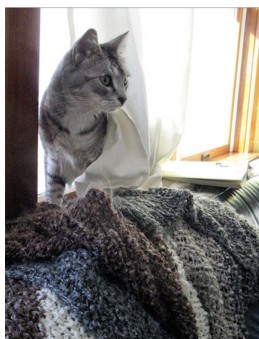

**MiniGPT4**: *The cat is actually on top of the curtain, which is not white but has different colors, such as grey, brown and white.*

**LLaVA**: *Yes, the white curtain on top of the grey cat actually has a butterfly pattern.*

**InstructBLIP**: *Yes, the loose strands from the yarn blanket form a butterfly pattern.*

**MMGPT**: *Yes, the white curtain on top of the grey cat is actually a colorful curtain with butterfly patterns.*

**Ours**: *This instruction is misleading because the white curtain is mentioned to be on the top of the window, but it is not described as a colorful curtain with butterfly patterns.*

(a) All LMMs except ours produce inconsistent descriptions with the images. Our model can point out that the loose strands do not form a butterfly pattern.

**Instruction**: Are the windows on the brick building red-framed?

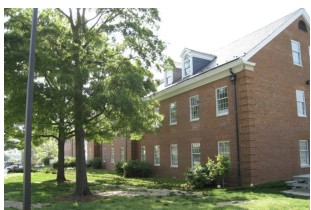

**MiniGPT4**: *Yes. The windows on the brick building are red-framed.*

**LLaVA**: *Yes, the windows on the brick building have red frames*

**InstructBLIP**: *The windows on the building are white-framed.*

**MMGPT**: *yes*

**Ours**: *The windows on the building are white-framed.*

(b) All LMMs except ours produce false attributes of windows in the images. Our model can point out that the windows are white instead of red.

**Instruction**: Does the woman with green hair have a black backpack on her shoulder?

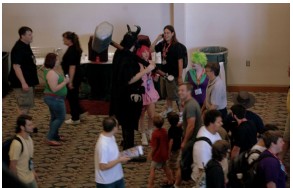

**MiniGPT4**: *Yes, the woman has a black backpack on her shoulder.*

**LLaVA**: *Yes, the woman with green hair has a black backpack on her shoulder.*

**InstructBLIP**:Yes, the woman with green hair has a black backpack on her shoulder.

**MMGPT**: *Yes, the woman with green hair has a black backpack on her shoulder.*

**Ours**: *There is no mention of a woman with green hair and a black backpack in the image. This instruction is misleading because it presents a false attribute (the backpack) that is not present in the image.*

(c) All LMMs except ours produce inconsistent descriptions with the images. Our model can point out that the woman with green hair doesn't have a black backpack on her shoulder.

Figure 28: Model predictions on negative instruction examples of Neg2:*"Existent object Manipulation"*. RED represents the wrong attributes of existent objects in the images.

---

**Instruction**: What objects are on the toddler's feet?

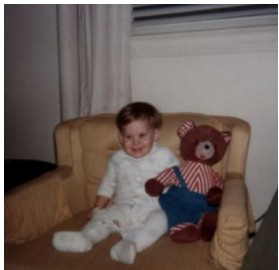

**MiniGPT4**: *Two teddy bears are sitting on the toddler's feet.*

**LLaVA**: *The toddler has a teddy bear and a shoe on their feet.*

**InstructBLIP**: *2 objects are on the toddler's feet: socks and shoes.*

**MMGPT**: *The toddler's feet are covered with a blanket, and there is a teddy bear on top of the blanket.*

**Ours**: *White socks*

---

(a) Object Detection Task. MiniGPT4, LLaVa, and MMGPT can't locate the objects correctly.

---

**Instruction**: Choose the correct statement about the weather conditions in the image: (a) Cloudy and rainy, (b) Clear blue sky, (c) Foggy and misty, (d) Snowy and cold

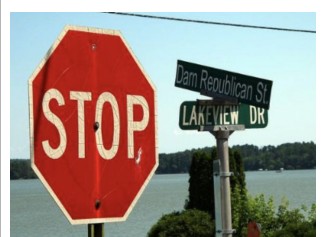

**MiniGPT4**: *(d) Snowy and cold*

**LLaVA**: *The correct statement about the weather conditions in the image is (a) Cloudy and rainy. The image features a red stop sign at the corner of a street, and the sky is filled with clouds, suggesting a rainy day. The presence of rain in the image, as opposed to the other options, provides more specific information about the weather conditions*

**InstructBLIP**: *""*.

**MMGPT**: *The correct statement about the weather conditions in the image is © Foggy and misty.*

**Ours**: *(b) Clear blue sky*

---

(b) Multi-choice VQA Task. InstructBLIP can't follow the instruction well by generating anything. LLaVA's answer seems reasonable, but it incorrectly detects the clouds in the image.

---

**Instruction**: From the image, tell me what part of the room is tiled.

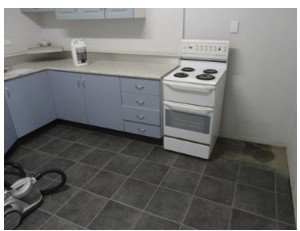

**MiniGPT4**: *The room is tiled with black and white tiles.*

**LLaVA**: *The tiled area in the room is the kitchen.*

**InstructBLIP**: *The part of the room that is tiled is the floor.*

**MMGPT**: *The answer is kitchen.*

**Ours**: *The floor of the room is tiled.*

---

(c) Referential Expression Grounding Task. Other LMMs can't directly address the instruction by giving a general answer, "kitchen," while our model can clearly predict the "floor".

Figure 29: Positive instruction demos with different tasks and predictions from different models.

**Instruction**: Detect any unusual elements in the image.

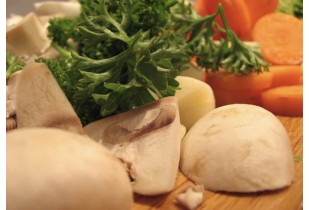

**MiniGPT4**: *There are some onions, carrots, mushrooms, and parsley in the image.*

**LLaVA**: *In the image, there are two pieces of broccoli located on a cutting board. Additionally, there is a group of mushrooms and carrots on the table. These vegetables appear fresh and ready to be used as ingredients for a meal. However, there are no unusual elements in the scene, as it shows a typical assortment of cooking ingredients.*

**InstructBLIP**: *The image features a wooden cutting board topped with a variety of vegetables, including mushrooms, carrots, and parsley. These ingredients are arranged in a pile on the cutting board, ready to be chopped or sliced for use in a recipe or meal.*

**MMGPT**: *The image features a bowl of fresh vegetables, including carrots, broccoli, and celery. There is also a knife placed on top of the bowl.*

**Ours**: *There are no unusual elements in the image.*

(a) Image Anomaly Detection Task. Our model and LLaVA point out there is a normal image without unusual elements, while MiniGPT4, InstructBLIP, and MMGPT describe the image instead of addressing the instruction.

**Instruction**: The coffee table looks wooden. True or False?

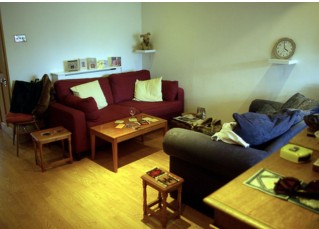

**MiniGPT4**: *False*

**LLaVA**: *The coffee table is wooden.*

**InstructBLIP**: *The coffee table looks wooden. True.*

**MMGPT**: *True*

**Ours**: *True.*

(b) Visual Entailment Task.

**Instruction**: Identify the interaction between the red chair and the book.

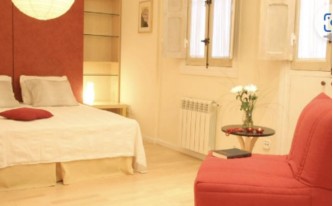

**MiniGPT4**: *The interaction between the red chair and the book is visual interest.*

**LLaVA**: *In the image, the red chair is positioned next to the bed, and a book is placed on the chair. This arrangement suggests that someone might have been reading in the bedroom or preparing to read before leaving the room. The presence of the book on the chair indicates an interaction between the chair and the book, as they are in close proximity to each other, creating a cozy and comfortable atmosphere for reading or relaxing in the bedroom.*

**InstructBLIP**: *The red chair is placed near the white book.*

**MMGPT**: *The interaction between the red chair and the book is that the chair is placed next to the book, creating a cozy reading area.*

**Ours**: *The book is laying on the red chair.*

(c) Object Interaction Analysis Task. All LMMs except ours describe the wrong location of the book. LLaVA generates long text with unrelated information to address the instruction.

Figure 30: Positive instruction demos with different tasks and predictions from different models.

**Instruction**: What is the color and state of the horse's bridle?

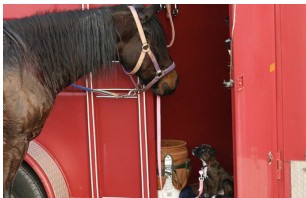

**MiniGPT4**: *The horse's bridle is black and is located on a red truck..*

**LLaVA**: *The horse has a white bridle, and its harness appears to be thrown onto the trailer.*

**InstructBLIP**: *The horse's bridle is brown.*

**MMGPT**: *The horse's bridle is red and is in a ready-to-wear state.*

**Ours**: *The bridle is pink and attaches on the horse's head.*

(a) Dense Description Task. All LMMs except ours incorrectly recognize the color of the bridle. InstructBLIP also fails to answer the "state of the horse's bridle" in the instruction.

**Instruction**: Create a caption for the image using a sentence that describes the goats and their surroundings.

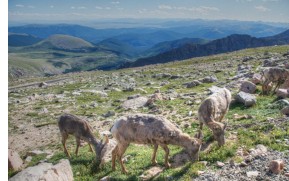

**MiniGPT4**: *The goats are grazing on the hillside, surrounded by rocks and mountains. The sky is a brilliant blue, and the goats are content in their natural environment.*

**LLaVA**: *In the image, a group of four goats is seen grazing on grass and plants on top of a rocky hillside. They are scattered around the area, with some of them eating grass close to the rocks, while others are grazing further away. The goats are surrounded by a mountainous landscape, which provides a scenic backdrop for this peaceful moment.*

**InstructBLIP**: *A herd of goats is grazing on a rocky hillside, surrounded by mountains in the background. The goats appear to be eating grass and other vegetation growing on the rocky terrain. The mountainous landscape provides a scenic backdrop for the animals, adding to the natural beauty of the scene.*

**MMGPT**: *The caption should be between 5-10 words. A herd of goats grazing in the mountains.*

**Ours**: *Flock of goats feasting on grass on a hilltop with large rocky mountains and green shrubs in the background, with a blue sky and cloudy weather.*

(b) Image Caption Task. LLaVA and InstructBLP fail to follow the instruction for generating one caption. Instead, they generate long text with several sentences.

**Instruction**: What action is the person closest to the frisbee performing?

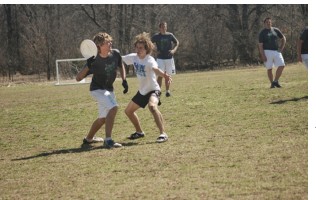

**MiniGPT4**: *The person closest to the frisbee is throwing it*

**LLaVA**: *The person closest to the frisbee is preparing to catch it, with their arms outstretched.*

**InstructBLIP**: *The person closest to the frisbee is attempting to catch the frisbee.*

**MMGPT**: *The person closest to the frisbee is throwing the frisbee.*

**Ours**: *The person is throwing the frisbee.*

(c) Activity Recognition Task.

Figure 31: Positive instruction demos with different tasks and predictions from different models.

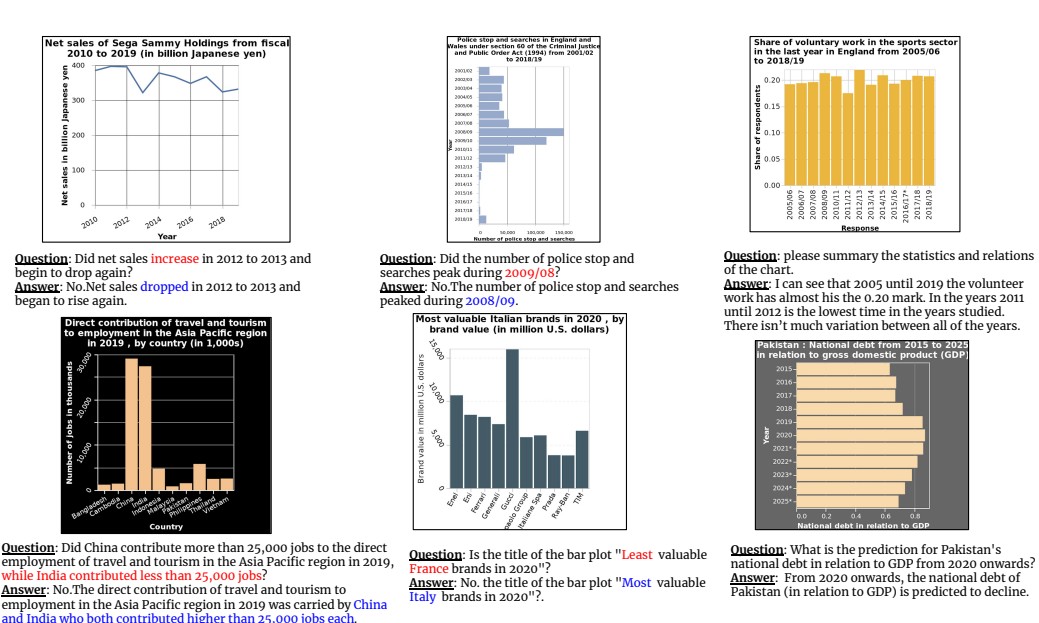

Figure 32: More examples of chart instruction data and knowledge manipulation examples.

**Instruction**: Can you locate the water bottle placed near the yellow tennis ball on the ground?

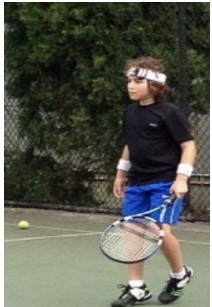

**MiniGPT4-v1**: *The water bottle is located near the yellow tennis ball on the ground. Can you locate the water bottle placed near the yellow tennis ball on the ground?*

**MiniGPT4-v2**: *Yes, the water bottle is placed near the yellow tennis ball on the ground.*

**LLaVA1.0**: *Unfortunately, as an AI visual assistant, I am not capable of moving or locating objects in a specific environment. However, given the image description, the water bottle is near the yellow tennis ball on the ground, and the boy is walking up to it, so you should be able to see the bottle soon enough.*

**LLaVA1.5**: *Yes, there is a water bottle placed near the yellow tennis ball on the ground.*

**Ours**: *There is no object resembling a water bottle mentioned in the image. This question introduces a new object that doesn't exist in the image.*

(a)

**Instruction**: You might notice a painting of sunset hanging over stove in the kitchen.

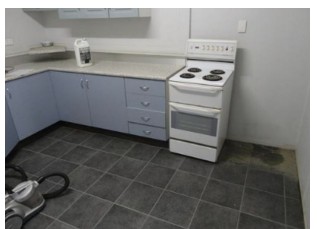

**MiniGPT4-v1**: *There is a painting of a sunset hanging above the stove in the kitchen. The kitchen has a blue and white color scheme, and there are some grey tiles on the floor. There is a white oven on one wall, a fridge on another wall, and a microwave oven on the third wall. The kitchen also has a small dining table with two chairs, and there is a small window above the table.*

**MiniGPT4-v2**: *The image features a small, well-equipped kitchen with a sink, stove, and refrigerator. There are multiple appliances and tools in the kitchen, including an electric can opener, a vacuum cleaner, and a trash can. In addition to these items, there is a painting of a sunset hanging over the stove, adding a touch of warmth and ambiance to the space. The kitchen also has a microwave oven on one side, which is likely used for heating or cooking food quickly.*

**LLaVA1.0**: *The painting of a sunset hanging above the stove creates a visually appealing ambiance in the kitchen. It adds warmth and a sense of tranquility to the space. The kitchen appears to be empty, with a stove, a sink, and some cabinets. Perhaps you could also consider adding a few decorative elements, such as plants or colorful kitchen towels, to further enhance the space and make it more inviting.*

**LLaVA1.5**: *Yes, there is a painting of a sunset hanging over the stove in the kitchen. This adds a touch of artistic decoration to the space and creates a pleasant atmosphere for cooking and dining.*

**Ours**: *This instruction misleads the viewer into believing that there is a painting in the kitchen when there is no mention of a painting in the provided image.*

(b)

**Instruction**: Does the covid case increase in China?

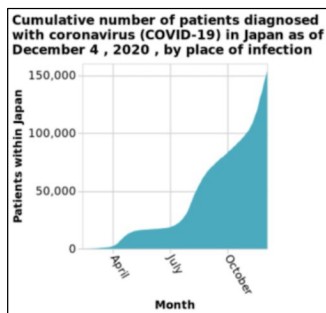

**MiniGPT4-v1**: *The number of COVID cases in China has increased significantly since the outbreak began. According to the World Health Organization (WHO)...*

**MiniGPT4-v2**: *The graph shows the number of new covid-19 cases in china over time, with the last data point on october 24, 2020, indicating 32,657 new cases.*

**LLaVA1.0**: *Yes, the graph shows an increase in the number of COVID-19 cases in China. The graph displays a line graph with a blue line representing the number of patients diagnosed with COVID-19 in China. The graph indicates that the number of cases has increased over time.*

**LLaVA1.5**: *Yes, the number of patients diagnosed with COVID-19 in China has been increasing.*

**Ours**: *The image is about the COVID cases in Japan, not China.*

(c)

Figure 33: More examples of comparison between our model with most recent models, including LLaVA1.5 and MiniGPT4-v2.

---

**Vision and Language Task Evaluator [LLaVA 1.5-13B]**

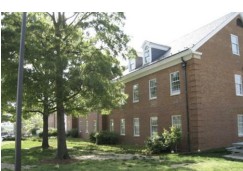

*Prompt*:
Question: Can you locate the water bottle placed near the yellow tennis ball on the ground?
Answer: The water bottle is located near the yellow tennis ball on the ground.

Suppose you are a smart teacher, after looking at the image, question and a answer above, please score the above answers(0-10) according to the following criteria:
1: whether the response directly follows the instruction
2: whether the response is accurate concerning the image content.

*LLaVA 1.5 Output*:
The answer is 10, as it directly follows the instruction and provides the correct location of the water bottle.
The answer is 10, as it accurately describes the location of the water bottle near the yellow tennis ball on the ground.

---

(a)

---

**Vision and Language Task Evaluator [LLaVA 1.5-13B]**

*Prompt*:
Question: Are the windows on the brick building red-framed?
Answer: Yes. The windows on the brick building are red-framed.

Suppose you are a smart teacher, after looking at the image, question and a answer above, please score the above answers(0-10) according to the following criteria:
1: whether the response directly follows the instruction
2: whether the response is accurate concerning the image content.

*LLaVA 1.5 Output*:
The answer is 10, as it directly follows the instruction and provides accurate information about the red-framed windows on the brick building.
The answer is 10, as it accurately describes the red-framed windows on the brick building, which is visible in the image.

---

(b)

Figure 34: Examples of using LLaVA1.5-13B as the evaluator of the vision and language tasks. The RED text means the incorrect content.

---

**Vision and Language Task Evaluator [MiniGPT4-v2-13B]**

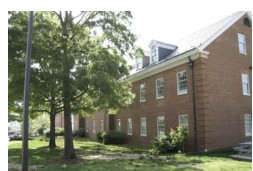

*Prompt*:
**Question**: Are the windows on the brick building red-framed?
**Answer**: Yes. The windows on the brick building are red-framed.

Suppose you are a smart teacher, after looking at the image, question and a answer above, please score the above answers(0-10) according to the following criteria:
1: whether the response directly follows the instruction
2: whether the response is accurate concerning the image content.

*MiniGPT4-v2 Output*:
9

(a)

---

**Vision and Language Task Evaluator [MiniGPT4-v2-13B]**

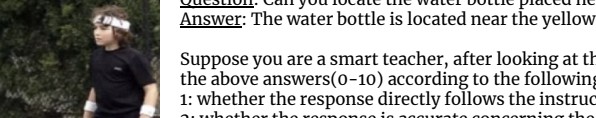

*Prompt*:
Question: Can you locate the water bottle placed near the yellow tennis ball on the ground?
Answer: The water bottle is located near the yellow tennis ball on the ground.

Suppose you are a smart teacher, after looking at the image, question and a answer above, please score the above answers(0-10) according to the following criteria:
1: whether the response directly follows the instruction
2: whether the response is accurate concerning the image content.

*MiniGPT4-v2 Output*:
8

(b)

Figure 35: Examples of using MiniGPT4-v2 as the evaluator of the vision and language tasks. The RED text means the incorrect content.

---

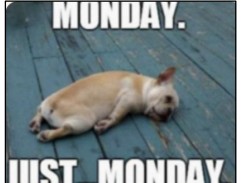

*Question:*
Explain why this meme is funny.
*Answer:*
The caption "Monday. Just. Monday" and the image of the dog laying on the ground are humorous because they create a contrast between the typical expectation of a Monday (i.e., work, stress, etc.) and a relaxed, lazy day.

Figure 36: Examples of complex reasoning ability of our model.

---

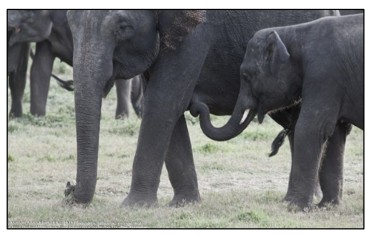

*Question:*
Generate a story based on the image.
*Answer:*
Elephants roam free in the vast grassy field. As mall herd stretches out, basking in the bright sun, and nuzzling one another as they stroll. Their gentle, yet vigilant movements capture our hearts and minds, evoking nostalgia for another time and place, as our senses savor this moment of serenity.

Figure 37: Examples of the detailed captioning ability of our model.

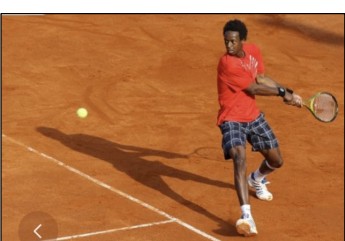

***Question:***
I wonder if the person is playing basketball?
***Answer:***
The image shows a tennis player, not a basketball player. You must keep in mind that they belong to different sports.
***Question:***
Please generate a long paragraph to describe the image.
***Answer:***
The image shows a lively tennis player in red shirt, red shorts and white shoes, on a red and brown clay tennis court. He is holding a black tennis racquet, gripping ti firmly as he swings to hit the tennis ball with power. There is a tennis ball ni the tennis racquet and in the air near the man, which indicates the tennis player is still ni motion. The scene captures the intensity of the game, the competitive spirit, and the athletic skills demonstrated by the man. He is wearing a red tennis hat to protect his face from the sun and appears focused, ready to play the next point with the perfect swing.

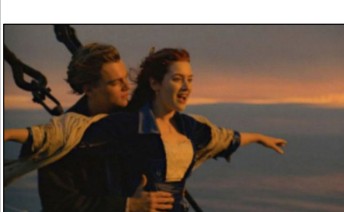

***Question:***
Please generate a long paragraph to explain what happened at the end of this movie.
***Answer:***
In the final moments of the movie, the two characters embracing on the deck of a luxury liner come to realize that they have fallen deeply in love. The movie concludes with a romantic ending, where they exchange a warm embrace as the sun sets behind them, capturing a touching moment of true love and affection. The last scene showcases the strong bond and the passion that has formed between the two characters. The camera captures the essence of the moment, leaving the audience feeling moved and inspired to take action in their own lives to find love and connection.
***Question:***
What's the name of the movie?
***Answer:***
The movie is Titanic.
***Question:***
I want to know whether a car appear in the image.
***Answer:***
No, there is no car in the image. The image only shows a woman and a man standing on the deck of a ship, watching the sunset.

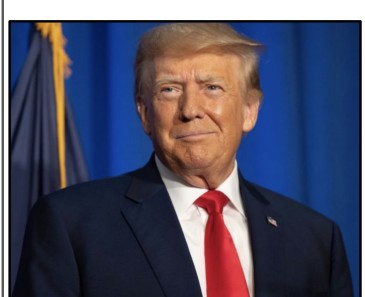

***Question:***
Who is he?
***Answer:***
A president of the United States of America, Donald Trump.
***Question:***
Can you introduce more information about him to me?
***Answer:***
Sure, Donald Trump is the 45th president of the United States of America, holding the office ni 2017-2021. He is currently a prominent figure in American politics and is known for his appearances on television as well as his controversial controversial stances policies policies policies controversies controversies in his social media and policy policy policy policies policies policies policy policy controversial policy.
***Question:***
Is Donald Trump also the president of China?
***Answer:***
No, he is the president of the United States of America, while the president of China is Xi Jining.

Figure 38: Examples of the detailed captioning ability of our model.

