# OpenReview forum: "Mitigating Hallucination in Large Multi-Modal Models via Robust Instruction Tuning"
_ICLR.cc/2024/Conference — ICLR 2024 poster_

### Official Review · Reviewer_VAPt · 2023-10-27

**Soundness:** 3 good
**Presentation:** 3 good
**Contribution:** 3 good
**Rating:** 8
**Confidence:** 3

**Summary:**

This work shows how the current Large Multimodal Models exhibit significant hallucinations in particular when presented with negative instructions. In this light, it introduces a new finetuning dataset, lRV-Instruction, of 400k visual instructions to fix this issue. This dataset covers 16 vision-language tasks and was generated using GPT-4. In particular, the authors use intermediate models/information as the visual inputs of GPT-4.

They also introduced a new evaluation benchmark, GAVIE, as a flexible approach to evaluate accuracy (i.e., hallucination in this case) and relevance (i.e., instruction following performance), without human intervention.  This dataset consists of 1000 humanly goldified pairs from the lRV-Instruction dataset. However, if the dataset is humanly goldified, it still reloes on a parametric model, namely GPT-4, to compute the scores.

**Strengths:**

The authors conducted a thorough analysis demonstrating the benefits of the lRV-Instruction dataset. For example, they show how finetuning on this dataset reduces models' hallucination (using in particular GAVIE) but also interestingly in three different independent tasks (MME, POPE, GQA). I find this result interesting and important. Also, they analyze better the benefit of the new dataset by looking at different aspects in their datasets (negative vs positive and tasks)

Furthermore, I find the generation process of lRV-Instruction interestingly simple as it leverages good existing LLMs and other visual models.

**Weaknesses:**

The authors conducted different experiments to show the feasibility of replacing the human annotator with GPT-4, both for constructing the dataset and for evaluation. If the dataset could be noisy (the authors do mention that the dataset is imperfect), the evaluation should be trusted and reproducible. I think the authors should outline further this limitation. Also, another useful experiment is to report the correlation of GPT-4 rating with human rating. My understanding is that the authors already have this dataset with expert raters but on a different scale. So such a correlation should be easy to report.

**Questions:**

1. For GAVIE the scores are 0-10. What is the intuition for such a big scale? My sense is that the task could be summarised to a 0/1 score. I like the analysis of the “Is GPT4-Assisted Evaluation Stable?”. Why not changing the benchmark to a (0-4) scores then?

I am also curious to know if the human experts used all the (0-4) scores provided. If only binary, you may just use a binary score too.


2. I don't understand the last sentence of this part
"Alternatively, (Li et al., 2023c; Fu et al., 2023) formulate the evaluation of hallucination as a binary classification task that prompts LMM to output "Yes" or "No". However, it is hard to evaluate the LMM output in an open-ended manner"

Why is the evaluation still challenging with a YES/NO question. This becomes just a classification task that does not need any human or parametric intervention.

Misc:
* Some bad formatting of citation and formulation in section “VISUAL INSTRUCTION TUNING"
* 3 incorrect examples in Figure 2 (neg). For example the question is about"Merkel" but the answer is about "Hilary Clinton"

---

> ### Author Response · Authors · 2023-11-21
>
> Thanks for the constructive feedback! We hope to clarify the confusion and answer the questions below. Feel free to let us know if you have any further questions!
>
> **Incorrect examples in Figure 2 (neg):**
>
> The question mentioned "Merkel" is not incorrect but a negative instruction/question. That is, "Merkel" doesn’t appear in the image. Therefore, the answer should be, "No, Hillary Clinton arrived at the Los Angeles Get Out The Vote Rally in the image.” We use this kind of negative samples to finetune Large Multimodal Models (LMMs), so that these models can learn to say “no” if the objects mentioned in the questions do not appear in the image, hence reducing hallucinations.
>
> **GAVIE Can Evaluate Open-ended Questions:**
>
> POPE [8] formulates the evaluation of hallucination as a binary classification task that prompts LMM to answer with "Yes" or "No".  However, if the question is a multiple choice question (e.g., “Choose the correct statement about the weather conditions in the image: (a) Cloudy and rainy, (b) Clear blue sky, (c) Foggy and misty, (d) Snowy and cold”) or  an open-ended question (e.g., “What objects are on the toddler’s feet?”), POPE can not evaluate model hallucinations on these questions as the answer space is much more complicated.  In contrast, our proposed GAVIE can evaluate model hallucinations even when the answers are open-ended manners.  (Fig 21-Fig 26 in the appendix)
>
> **GAVIE Aligns well with Human Experts**
>
> To assess the correlation of GPT-4 rating with the human rating, we hired three experts specializing in NLP to perform the human assessment. From Tab. 4 in the paper, all experts agree that the output from our model is the best, followed by InstructBLIP in the second place, and MMGPT performs the worst. This ranking aligns with our GAVIE. More details are shown in the A.1.1 (appendix).
>
> **Why GAVIE Uses Scores 0-10 instead of 0/1.**
>
> In an open-ended setting, the model’s answer to a question may be completely wrong, partly wrong, or entirely correct. With a score of 0-10, GAVIE can evaluate the model’s answer and produce a detailed level of correctness instead of only 0/1(correct/wrong), which does not give differentiable scores to answers that are partly wrong or completely wrong (i.e., both will receive a score of 0). However, our GAVIE can assign a score of 0 for the completely wrong answer and a partial score (e.g., 5) for the partly wrong answer. Therefore,  GAVIE can not only give a more accurate ranking among compared models but also help people understand the level of correctness for answers from different models. The relationship between correctness and scores in GAVIE is summarized as follows:
>
> RELEVANCY has four grade levels: (1) The response is entirely relevant (9-10), (2) The response is mostly relevant (6-8), (3) The response is partly relevant (3-5), (4) The response is seldom relevant (0-2). ACCURACY has four grade levels: (1) The response is completely accurate (9-10), (2) The response has minor errors (6-8), (3) The response is partly accurate (3-5), (4) The response is mostly or completely wrong (0-2).
>
> **Reference**
>
> [1] Liu, Haotian, et al. "Improved baselines with visual instruction tuning." arXiv preprint arXiv:2310.03744 (2023).
>
> [2] Chen, Jun, et al. "MiniGPT-v2: large language model as a unified interface for vision-language multi-task learning." arXiv preprint arXiv:2310.09478 (2023).
>
> [3] Liu, Fuxiao, et al. "Hallusionbench: You see what you think? or you think what you see? an image-context reasoning benchmark challenging for gpt-4v (ision), llava-1.5, and other multi-modality models." arXiv preprint arXiv:2310.14566 (2023).
>
> [4] Wang, Junyang, et al. "An LLM-free Multi-dimensional Benchmark for MLLMs Hallucination Evaluation." arXiv preprint arXiv:2311.07397 (2023).
>
> [5] Liu, Yang, et al. "Gpteval: Nlg evaluation using gpt-4 with better human alignment." arXiv preprint arXiv:2303.16634 (2023).
>
> [6] Kirillov, Alexander, et al. "Segment anything." arXiv preprint arXiv:2304.02643 (2023)
>
> [7] Wu, Jialian, et al. "Grit: A generative region-to-text transformer for object understanding." arXiv preprint arXiv:2212.00280 (2022).
>
> [8] Li, Yifan, et al. "Evaluating object hallucination in large vision-language models." arXiv preprint arXiv:2305.10355 (2023).

---

### Official Review · Reviewer_WNxt · 2023-10-30

**Soundness:** 4 excellent
**Presentation:** 3 good
**Contribution:** 4 excellent
**Rating:** 8
**Confidence:** 4

**Summary:**

Hallucination presents a substantial challenge in contemporary large multi-modal models (LMMs). To tackle this issue of hallucination, this paper introduces the LRV-Instruction dataset, encompassing 400k samples of instruction-following scenarios spanning 16 different vision-language tasks. The dataset is enriched with a substantial amount of negative instruction data, generated through three distinct heuristics: Manipulation of Nonexistent Objects, Manipulation of Existent Objects, and Manipulation of Knowledge within instructions. Additionally, the paper proposes GAVIE, an innovative methodology for the automatic evaluation of LMMs. GAVIE assesses models based on two critical dimensions: accuracy and relevance. Empirical results from the experiments reveal that applying instruction tuning to MiniGPT4 and mplug-owl significantly enhances their performance, surpassing their original instruction-tuning results on numerous established public datasets.

**Strengths:**

The paper presents LRV-Instruction, a substantial and varied benchmark tailored for visual instructions. It covers 16 vision-and-language tasks and includes both positive and negative instructions, making it more robust and comprehensive compared to existing datasets.

The proposed GAVIE allows for the assessment of LMMs' hallucination without the need for human-annotated groundtruth answers. This method substantially increases the efficiency and scalability of assessing adjustments made to visual instructions, with human validation lending further support.

The empirical investigation underscores the utility of LRV-Instruction in diminishing hallucination and augmenting the performance of models. The finetuning results achieving state-of-the-art performance on both the evaluation set and public benchmarks, add credibility to the proposed approach.

**Weaknesses:**

Even though the LRV-Instruction is generated automatically via GPT-4, its dependence on the VG dataset's high-quality annotations, in my opinion, may pose a constraint on its ability to scale. Also I think more analyses on more scenarios of LLMs should be included into discussions.

**Questions:**

N/A

---

> ### Author Response · Authors · 2023-11-21
>
> Thanks for the constructive feedback! We hope to clarify the confusion and answer the questions below. Feel free to let us know if you have any further questions!
>
> **Use Pseudo Data to Scale-up**
>
> To demonstrate the scalability of our dataset, we use GRiT [7]  to generate pseudo-dense captions in place of the high-quality annotations from humans in the Visual Genome dataset. From Tab. 5, we found finetuning on pseudo captions can also improve the model performance. This demonstrates that our visual instruction generation method has great potential to be further scaled up, even when without groundtruth dense captions. Please refer to Section 6.3 for more details.
>
> **Reference:**
>
> [1] Liu, Haotian, et al. "Improved baselines with visual instruction tuning." arXiv preprint arXiv:2310.03744 (2023).
>
> [2] Chen, Jun, et al. "MiniGPT-v2: large language model as a unified interface for vision-language multi-task learning." arXiv preprint arXiv:2310.09478 (2023).
>
> [3] Liu, Fuxiao, et al. "Hallusionbench: You see what you think? or you think what you see? an image-context reasoning benchmark challenging for gpt-4v (ision), llava-1.5, and other multi-modality models." arXiv preprint arXiv:2310.14566 (2023).
>
> [4] Wang, Junyang, et al. "An LLM-free Multi-dimensional Benchmark for MLLMs Hallucination Evaluation." arXiv preprint arXiv:2311.07397 (2023).
>
> [5] Liu, Yang, et al. "Gpteval: Nlg evaluation using gpt-4 with better human alignment." arXiv preprint arXiv:2303.16634 (2023).
>
> [6] Kirillov, Alexander, et al. "Segment anything." arXiv preprint arXiv:2304.02643 (2023)
>
> [7] Wu, Jialian, et al. "Grit: A generative region-to-text transformer for object understanding." arXiv preprint arXiv:2212.00280 (2022).
>
> [8] Li, Yifan, et al. "Evaluating object hallucination in large vision-language models." arXiv preprint arXiv:2305.10355 (2023).

---

> > ### Comment · Reviewer_WNxt · 2023-11-22
> >
> > I appreciate the author's detailed responses to the review comments. Having reviewed the perspectives of other reviewers and the author's comprehensive answers, I believe this paper has a significant positive impact on the field of MLLM. I encourage the author to further refine the paper based on feedback from all reviewers. My initial score for this submission remains unchanged.

---

### Official Review · Reviewer_LKUy · 2023-10-31

**Soundness:** 3 good
**Presentation:** 3 good
**Contribution:** 3 good
**Rating:** 8
**Confidence:** 3

**Summary:**

- Vision and Language multimodal LLM are known to be prone to hallucination, especially for "negative (non-existence)" references.
- One possible reason is the existing instruction fine-tuning basically relies on massive "positive" instructions.
- The manuscript first introduces a new dataset that contains a variety of "negative" instruction cases.
- At the same time, the manuscript proposes the use of GPT-4 for automatic evaluations of open-ended questions.
- Experimental results indicate the efficacy of the new dataset for mitigating the multi-modal LLM hallucinations

After author feedback, score is upgraded.

**Strengths:**

- For the reviewer, this is the first proposal of the "balanced" instruction fine-tuning dataset for multi-modal LLM. Motivations for such dataset are reasonable.
- (Class-) balances of the training corpus are fatal for successful model training. In that sense, it is reasonable that the proposed dataset can mitigate the hallucinations of the LLMs.
- it is good to know that (strong) LLM can help the open-ended questions in V-L: checking the images and the answers are time-consuming.

**Weaknesses:**

- As cited in the manuscript, (Liu 2023d) proposed one of the first GPT-based automatic evaluations without human intervention. The manuscript does not explain the main differences from (Liu 2023d), except the application to vision - language domain.
- GAVIE relies on the black-box GPT-4 engine. It means the evaluation is affected by the system update of the GPT-4, implying we cannot assure the loyal reproducing of GAVIE results in months/years after. Are there any remedies for this problem?

**Questions:**

- Concerning the 2nd point of the "weakness", I wonder what happens to GAVIE if we replace the engine from GPT-4 to (possibly weaker) multi-modal LLMs.

---

> ### Author Response · Authors · 2023-11-21
>
> Thanks for the constructive feedback! We hope to clarify the confusion and answer the questions below. Feel free to let us know if you have any further questions!
>
> **Difference with G-Eval [5]**
>
> The difference between our evaluation method (GAVIE) and G-eval is as follows:
>
> (1) Unlike G-Eval [5], which mainly focuses on text summarization and dialogue tasks, our GAVIE method can adapt to diverse instruction formats and tasks. To demonstrate the effectiveness of GAVIE, we compare GAVIE and human expert evaluation on our evaluation set, which includes 16 vision and language (VL) tasks. From Tab 4, we found that GAVIE aligns well with human evaluation.
>
> (2)Unlike G-Eval [5], which only generates one score for each sample according to its evaluation form, our GAVIE generates different scores for different criteria:  (a) Relevancy: whether the response directly follows the instruction; (b) Accuracy: whether the response is accurate concerning the image content. Additionally, GAVIE also provides explanations to support the scores. In this way, people can clearly understand how the models perform in terms of different criteria and why GAVIE gives this score.
>
> **Blackbox GPT-4:**
>
> To reproduce the results of GAVIE, we indicate the GPT4 version ( GPT4-32k-0314) we used in Section 5. We also show some examples in Fig 34 and Fig 35  by replacing GPT4 with current popular open-sourced Large Multimodal Models (LMMs), including LLaVA1.5-13B [1] and MiniGPT4-v2 [2]. From Fig 34, we can see that LLaVA1.5-13B can follow the instructions to evaluate the answers according to two criteria, but it fails to detect that the answers are incorrect, given the image content. From Fig 35, we can see that MiniGPT4-v2-13B can not follow the instructions to generate a score for each criterion.  What’s more, the score MiniGPT4-v2-13B  generates is not reasonable. Therefore, current open-sourced LMMs like LaVA1.5-13B and MiniGPT4-v2 may struggle when evaluating hallucination in vision-and-language tasks.
>
> We come up with two possible remedies, which are promising directions to explore in future works:
>
> (1) One can finetune LLaVA1.5 on the vision and language evaluation tasks to enhance its instruction following abilities. Additionally, to further enhance LLaVA1.5, one can also replace the vision encoders with [6] to improve its visual perception ability to make it a better evaluator.
>
> (2) One can explore LLM-free methods for VL hallucination evaluation. For example,  we can design questions with answers from a closed set (e.g., binary classification) based on different types of hallucinations. For object hallucination, we can construct the question as “Is there a {hallucination object} in this image?”.  For attribute hallucination, we can design questions like “​​Is the {object} {state} in this image?”, “Does the {object} {action} in this image?”, “Is the {object} {color} in this image?”. For relation hallucination, we can construct questions like “Is there a direct contact between the {object 1} and {object 2} in this image?”.  For all the cases described above, the groundtruth answers to these questions will be “yes” or “no”. In this way, we can evaluate the output of LMMs without GPT4.
>
>
> **Reference:**
>
> [1] Liu, Haotian, et al. "Improved baselines with visual instruction tuning." arXiv preprint arXiv:2310.03744 (2023).
>
> [2] Chen, Jun, et al. "MiniGPT-v2: large language model as a unified interface for vision-language multi-task learning." arXiv preprint arXiv:2310.09478 (2023).
>
> [3] Liu, Fuxiao, et al. "Hallusionbench: You see what you think? or you think what you see? an image-context reasoning benchmark challenging for gpt-4v (ision), llava-1.5, and other multi-modality models." arXiv preprint arXiv:2310.14566 (2023).
>
> [4] Wang, Junyang, et al. "An LLM-free Multi-dimensional Benchmark for MLLMs Hallucination Evaluation." arXiv preprint arXiv:2311.07397 (2023).
>
> [5] Liu, Yang, et al. "Gpteval: Nlg evaluation using gpt-4 with better human alignment." arXiv preprint arXiv:2303.16634 (2023).
>
> [6] Kirillov, Alexander, et al. "Segment anything." arXiv preprint arXiv:2304.02643 (2023)
>
> [7] Wu, Jialian, et al. "Grit: A generative region-to-text transformer for object understanding." arXiv preprint arXiv:2212.00280 (2022).
>
> [8] Li, Yifan, et al. "Evaluating object hallucination in large vision-language models." arXiv preprint arXiv:2305.10355 (2023).

---

> > ### Comment · Reviewer_LKUy · 2023-11-21
> > **Thank you authors**
> >
> > Thank you for providing kind answers to my questions.
> > I'm still struggling to read all the review comments and answers. I will post some comments whenever I encounter additional questions and comments.

---

> > > ### Author Response · Authors · 2023-11-21
> > >
> > > Dear Reviewer:
> > >
> > > Thanks for your response! If you have any further concerns and questions, please do not hesitate to let us know. We will be happy to address them promptly.
> > >
> > > Best Regards.

---

> > > > ### Comment · Reviewer_LKUy · 2023-11-23
> > > > **I will change score upward**
> > > >
> > > > Given the feedbacks from fellow reviewers and the authors, now I'm positive for acceptance.
> > > >
> > > > The current scheme heavily relies on a specific version of GPT-4, this is the only remaining concern.
> > > > But in the other parts I find no fatal drawbacks.
> > > > I will upgrade to 8.

---

### Official Review · Reviewer_68kD · 2023-11-03

**Soundness:** 3 good
**Presentation:** 2 fair
**Contribution:** 3 good
**Rating:** 6
**Confidence:** 4

**Summary:**

The paper introduces the Large-scale Robust Visual (LRV)-Instruction dataset with both positive and negative visual instructions. The authors use GPT4-Assisted Visual Instruction Evaluation (GAVIE) as the evaluation tool to evaluate hallucinations. The authors finetuned the popular baselines on the proposed LRV-instruction datasets and improved the hallucinations as well as the other commonly used benchmarks.

**Strengths:**

The motivation is good, especially for creating negative instructions to balance the datasets. The experiments are also very sufficient and compact, which is really appreciated by me. The paper writing is also good.

**Weaknesses:**

- Based on the claim "As observed by (Li et al., 2023c), current LMMs tend to answer "Yes" for any instructions presented to the model, even when the proper answer should be "No". Our investigation reveals that most LMMs are finetuned on unbalanced datasets containing only positive instructions " in Instruction, the author assumed current LLMs tend to answer "Yes" issues caused by unbalanced". This statement got confused because, so far LLaVA1.5 boosts the model without balancing positive and negative instructions. Also, before LLaVA1.5, even in LLaVA1.3 or 1.1 the model seems to not have this "always yes" issue anymore. It will be really helpful if the author can provide more experiments to prove these statements.
- It's not fair to compare models not trained on VG data with your own evaluation set created from VG. Please add more comparisons with models trained on VG data. Also, it would be great if the author could evaluate the current minigpt4-v2, LLaVA1.5. If the author thinks these two works are too recent, the author can also try Shikra, which also includes VG in the training set.
- One of the evaluations is "Relevancy: whether the response directly follows the instruction. " This actually evaluates the instruction following ability, which is kind of tricky, cause it's hard to say if the training data difference brings instruction domain difference, which makes the evaluation not fair.
- Human label with 4 scores. It would be great to give some explanation on what kind of aspect humans evaluate and preferred means it will be better if the author could provide a more specific evaluation policy and rules.

**Questions:**

How about the performance of generating complex reasoning and detailed captions after training the LVLM on LRV-instruction datasets? Although Llava data may have hallucinations, complex reasoning and detailed captions are still good, does the LRV dataset maintain the capabilities? Can you show us some examples of complex reasoning and detail captions?

For the other questions please refer to the weaknesses. Basically, the contribution of the dataset is good but we should be careful, I would like to hear from the authors about the concerns I bring up.

---

> ### Author Response · Authors · 2023-11-21
>
> Thanks for the constructive feedback! We hope to clarify the confusion and answer the questions below. Feel free to let us know if you have any further questions!
>
> **Compare Our Model with LLaVA1.5 and other LMMs:**
>
> As current open-source LMMs are built on strong LLMs, they may over-rely on language priors (parametric memory) and generate words more likely to go together with the instruction text regardless of the image content. To address the hallucination, besides the positive instructions with 16 tasks, we build negative instructions with existent object (attributes and relationship) manipulation, non-existent object manipulation, and knowledge manipulation. After finetuning LMMs on our dataset, their answers will be more consistent with the image.
>
> Following the suggestions by the reviewer, we compare LLaVA1.5 with our model across three datasets. Note that the backbone of our model is Mplug-owl.
>
> (1) We first analyze the discriminative task of  AMBER [4], including object existence, object attribute, and object relation hallucination. The evaluation score is F1, and the groundtruth answer is yes or no.
> |  Method   | Existence  | Attribute | Relation |
> |  ----  | ----  |---  |---  |
> | mPLUG-Owl-7B | 29.9% |33.5% |26.4% |
> | MiniGPT4-v2-7B [2] | 80.0% |41.1% |57.9% |
> | LLaVA1.5-7B [1] | 83.2% |64.0% |65.7% |
> | Ours-7B | 81.4% |70.4% |69.2% |
>
> Although our model achieves a slightly lower F1 score regarding object existence, we outperform LLaVA1.5 in both the object attribute and object relation hallucinations. We attribute our model’s success to the existent object (attributes and relationship) manipulation in the negative instructions.
>
> (2) We secondly analyze the results of our evaluation set via GAVIE. Both LLaVA1.5 and MiniGPT-v2 use Visual Genome as their training data.
> |  Method   | GAVIE-Accuracy  | GAVIE-Relevancy |
> |  ----  | ----  |---  |
> | LLaVA1 -7B | 4.36 |6.11 |
> | LLaVA 1.5-7B [1] | 6.42 |8.20 |
> | MiniGPT4-v1-7B | 4.14 |5.81 |
> | MiniGPT4-v2-7B [2] | 6.01 |8.10 |
> | mPLUG-Owl-7B | 4.84 |6.35 |
> | Ours-7B | 6.58 |8.46 |
>
> Although LLaVA1.5 and MiniGPT4-v2 improve significantly compared with LLaVA1 and MiniGPT4-v1, our model still achieves the best accuracy and relevancy.  We further show some qualitative examples in Fig. 33 of the appendix.
>
> (3) We finally analyze Hallusionbench [3], an image-context reasoning benchmark covering various topics and image types. The groundtruth answer is yes or no.
> |  Method   | Accuracy (gt=yes)  | Accuracy (gt=no) |
> |  ----  | ----  |---  |
> | LLaVA 1.5-7B [1] | 92% |15% |
> | MiniGPT4-v2-7B [2] | 85% |11% |
> | Ours-7B | 88% |47% |
>
> From the results, we find that both LLaVA1.5 and MiniGPT4-v2 achieve high accuracy on the positive set but perform poorly on the negative set. Our model can achieve the same level of accuracy when groundtruth answer is yes and much higher accuracy when groundtruth answer is no. We attribute our model’s success to the knowledge manipulation in the negative instructions.
>
> Generally speaking, LLaVA 1.5 performs well when meeting object hallucinations. However, attribute hallucination, relation hallucination, and knowledge hallucination are still challenging for LLaVA 1.5. Therefore,  our LRV-Instruction dataset with various manipulation negative instructions and positive instructions of 16 tasks is an excellent resource to address the hallucinations.
> Thank you for your comments and suggestions. We will include the discussions in our final paper.
>
> **Relevancy Score:**
>
> To make a fair comparison, we compare our model with LLaVA1.5 and MiniGPT4-v2, which both use the Visual Genome as their training data to address the domain difference.  From our evaluation set (GAVIE),  our model outperforms both LLaVA1.5-7B and MiniGPT4-v2-7B regarding the instruction-following ability.
> |  Method   | GAVIE-Relevancy  |
> |  ----  | ----  |
> | LLaVA 1.5-7B [1] | 8.20 |
> | MiniGPT4-v2-7B [2] | 8.10 |
> | Ours-7B | 8.46 |
>
> **Human Evaluation Policy:**
>
> The details of human evaluation guidelines are shown below:
>
> As for each question, there are an instruction, an image, and several answers. Suppose you are a smart teacher, please score the answers according to the two criteria. (1) Accuracy: whether the response is accurate concerning the image content. (2) Relevancy: whether the response directly follows the instruction without unrelated answers. There are four options for the scores:
> A score of 1 ( Very Poor): The response is seldom relevant to the instruction and is mostly wrong according to the image.
> A score of 2 ( Poor): The response is partly relevant to the instruction and partly accurate according to the image.
> A score of 3 (Good): The response is mostly relevant to the instruction and has minor errors according to the image.
> A score of 4 (Excellent): The response is completely relevant to the instruction and completely accurate according to the image.

---

> > ### Author Response · Authors · 2023-11-21
> >
> > **Complex reasoning and detailed captions Examples:**
> >
> > Some examples are shown in Fig 36, Fig 37, and Fig 38 (appendix) to show our complex reasoning and detailed captioning ability.
> >
> > **Reference:**
> >
> > [1] Liu, Haotian, et al. "Improved baselines with visual instruction tuning." arXiv preprint arXiv:2310.03744 (2023).
> >
> > [2] Chen, Jun, et al. "MiniGPT-v2: large language model as a unified interface for vision-language multi-task learning." arXiv preprint arXiv:2310.09478 (2023).
> >
> > [3] Liu, Fuxiao, et al. "Hallusionbench: You see what you think? or you think what you see? an image-context reasoning benchmark challenging for gpt-4v (ision), llava-1.5, and other multi-modality models." arXiv preprint arXiv:2310.14566 (2023).
> >
> > [4] Wang, Junyang, et al. "An LLM-free Multi-dimensional Benchmark for MLLMs Hallucination Evaluation." arXiv preprint arXiv:2311.07397 (2023).
> >
> > [5] Liu, Yang, et al. "Gpteval: Nlg evaluation using gpt-4 with better human alignment." arXiv preprint arXiv:2303.16634 (2023).
> >
> > [6] Kirillov, Alexander, et al. "Segment anything." arXiv preprint arXiv:2304.02643 (2023)
> >
> > [7] Wu, Jialian, et al. "Grit: A generative region-to-text transformer for object understanding." arXiv preprint arXiv:2212.00280 (2022).
> >
> > [8] Li, Yifan, et al. "Evaluating object hallucination in large vision-language models." arXiv preprint arXiv:2305.10355 (2023).

---

> > > ### Author Response · Authors · 2023-11-21
> > >
> > > Dear Reviewer:
> > >
> > > We would like to again express our sincere appreciation for your efforts and valuable feedback on our paper. We kindly hope that you can take some time to re-evaluate our paper based on our replies above. If you have any further concerns and questions, please do not hesitate to let us know. We will be happy to address them promptly.
> > >
> > > Best Regards.

---

> > > > ### Comment · Reviewer_68kD · 2023-11-23
> > > > **Respond**
> > > >
> > > > I sincerely appreciate the author's detailed reply and thorough experiments! My main concerns have been addressed. I will raise my score to 6. I agree with the other reviewer that this work has a positive contribution to this field. But I would like to highlight that the authors are a little bit overclaiming the contribution, which is also the reason that I did not raise to 8. I think this work addresses "yes or no" hallucination instead of a general hallucination problem, e.g., hallucination in captions. The authors should carefully claim them in the paper.

---

### Author Response · Authors · 2023-11-21

We thank all reviewers for the insightful feedback and encouraging comments. All reviewers highlighted the contributions of our paper:

(1) Reviewer 68kD and LKUy underscore that our motivation is good, especially for the first proposal of the “balanced” instruction dataset (LRV-Instruction) for LMM to mitigate hallucinations. LRV-Instruction includes 400k samples of instruction-following scenarios spanning 16 different vision-language tasks. It also has a substantial amount of negative instruction data generated through three distinct heuristics.

(2) Reviewer VAPt mentioned that he/she finds the generation process of lRV-Instruction interestingly simple as it leverages good existing LLMs and other visual models.

(3) Reviewer WNet mentioned that GAVIE allows for the assessment of LMMs' hallucinations without the need for human-annotated groundtruth answers. This method substantially increases the efficiency and scalability of assessing adjustments made to visual instructions.

(4) All four reviewers highlight the thorough analysis demonstrating the benefits of the lRV-Instruction dataset to reduce models' hallucinations in three different independent tasks (MME, POPE, GQA). We also run additional experiments by comparing our model with LLaVA1.5 [1] and MiniGPT4-v2 [2] on AMBER [4] and Hallusionbench [3]. The results demonstrate the effectiveness of our dataset.


We will also keep improving this work based on reviewers’ suggestions. The following references have been used in this rebuttal.

**Reference**

[1] Liu, Haotian, et al. "Improved baselines with visual instruction tuning." arXiv preprint arXiv:2310.03744 (2023).

[2] Chen, Jun, et al. "MiniGPT-v2: large language model as a unified interface for vision-language multi-task learning." arXiv preprint arXiv:2310.09478 (2023).

[3] Liu, Fuxiao, et al. "Hallusionbench: You see what you think? or you think what you see? an image-context reasoning benchmark challenging for gpt-4v (ision), llava-1.5, and other multi-modality models." arXiv preprint arXiv:2310.14566 (2023).

[4] Wang, Junyang, et al. "An LLM-free Multi-dimensional Benchmark for MLLMs Hallucination Evaluation." arXiv preprint arXiv:2311.07397 (2023).

[5] Liu, Yang, et al. "Gpteval: Nlg evaluation using gpt-4 with better human alignment." arXiv preprint arXiv:2303.16634 (2023).

[6] Kirillov, Alexander, et al. "Segment anything." arXiv preprint arXiv:2304.02643 (2023)

[7] Wu, Jialian, et al. "Grit: A generative region-to-text transformer for object understanding." arXiv preprint arXiv:2212.00280 (2022).

[8] Li, Yifan, et al. "Evaluating object hallucination in large vision-language models." arXiv preprint arXiv:2305.10355 (2023).

---

### Meta-Review · Area_Chair_hzXK · 2023-12-10

**Metareview:**

This paper presents the Large-scale Robust Visual (LRV)-Instruction dataset, a novel resource featuring both positive and negative visual instructions. The work has undergone thorough evaluation by four experts in the field.

The reviewers have collectively recognized the significance of this work, particularly as it represents the first proposal of a 'balanced' instruction fine-tuning dataset for multi-modal Large Language Models (LLMs). They have noted the clear and reasonable motivations behind the creation of such a dataset, underscoring its potential value to the field.

Reflecting on the insightful feedback provided by the reviewers, I am pleased to recommend this paper for acceptance at ICLR 2024. This decision highlights the paper's innovative approach and its potential impact on the research community in the area of multi-modal LLMs.
The authors are also strongly encouraged to address remaining comments from all the reviewers to the best of their ability. Congratulations again to the authors on the acceptance of their paper.

**Justification For Why Not Higher Score:**

This work addresses "yes or no" hallucination instead of a general hallucination problem, e.g., hallucination in captions.

**Justification For Why Not Lower Score:**

The authors have conducted a thorough analysis demonstrating the benefits of the lRV-Instruction dataset.

---

### Decision · Program_Chairs · 2024-01-16

Accept (poster)